



# Global database and model on dissolved carbon in soil solution

Joep Langeveld[1], Alexander F. Bouwman[1,2,3], Wim Joost van Hoek[1], Lauriane Vilmin[1], Arthur H. W. Beusen[1,2], José M. Mogollón[1,4], and Jack J. Middelburg[1]

[1]Department of Earth Sciences - Geochemistry, Faculty of Geosciences, Utrecht University, PO Box 80021, 3508TA Utrecht, the Netherlands.
[2]PBL Netherlands Environmental Assessment Agency, P.O. Box 30314, 2500GH the Hague, the Netherlands.
[3]Laboratory of Marine Chemistry Theory and Technology, Ministry of Education, Ocean University of China, Qingdao 266100, PR China.
[4]Department of Industrial Ecology, Leiden University, P.O. Box 9518, 2300RA Leiden, the Netherlands.

**Correspondence:** Joep Langeveld (j.j.langeveld@uu.nl)

**Abstract.** Dissolved carbon leaching in and from soils plays an important role in C transport along the terrestrial-aquatic continuum. However, a global overview and analysis of dissolved carbon in soil solutions, covering a wide range of vegetation types and climates, is lacking. We compiled a global database on annual average dissolved organic carbon (DOC) and dissolved inorganic carbon (DIC) in soil solutions, including potential governing factors, with 762 entries from 351 different sites

covering a range of climate zones, land cover types and soil classes. Using this database we develop regression models to calculate topsoil concentrations, and concentrations vs. depth in the subsoil at the global scale. For DIC, the lack of a proportional globally distributed cover inhibits analysis on a global scale. For DOC, annual average concentrations range from 1.7 to 88.3 (median=25.27) mg C/L for topsoils and from 0.42 to 372.1 (median=5.50) mg C/L for subsoils (excluding lab incubations). Highest topsoil values occur in forests of cooler, humid zones. In topsoils, multiple regression showed that precipitation is the

most significant factor. Our global topsoil DOC model ($R^2$=0.36) uses precipitation, soil class, climate zone and land cover type as model factors. Our global subsoil model describes DOC concentrations vs. depth for different USDA soil classes (overall $R^2$=0.45). Highest subsoil concentrations are calculated for Histosols.

## 1 Introduction

Freshwaters are important active systems in the global carbon (C) cycle, linking the land and the ocean (Cole et al., 2007).

Global inland waters are estimated to process about 2.9 Pg C/y (Battin et al., 2009; Regnier et al., 2013), but the magnitude of the C flux through streams, rivers, lakes and reservoirs is fraught with uncertainties (Aufdenkampe et al., 2011; Ciais et al., 2014). This C flux comprises litterfall from vegetation in floodplains and riparian zones of primarily headwaters, surface runoff, leaching from soils and processing via groundwater to surface waters (Schiff et al., 1990). In budget studies the groundwater C flux is commonly ignored (Clark et al., 2001; Kindler et al., 2011; Luyssaert et al., 2010; Don and Schulze, 2008) and while this

could lead to an inaccurate quantification of the terrestrial C budget (Kling et al., 1991; Luyssaert et al., 2010), several studies show that leaching and transport through groundwater of dissolved C is a major C source to surface waters (Marín-Spiotta et al., 2014; Hotchkiss et al., 2015; Winterdahl et al., 2016; Rasilo et al., 2017).





Leaching of C from soils occurs in the form of dissolved organic carbon (DOC) and dissolved inorganic carbon (DIC) (Schiff et al., 1990). DOC originates from direct leaching of organic matter, decomposition, or desorption from soil particles (Kalbitz et al., 2000). Dissolved $CO_2$ and related species bicarbonate and carbonate, together making up DIC, in the upper horizons or topsoil are constrained by the local atmospheric $CO_2$ concentration and soil pH. DIC stems from weathering, $CO_2$ dissolution

and respiration by plants, animals and microbes using DOC and particulate organic C (Shibata et al., 2001). A range of factors has been identified in studies as potential drivers of DOC and DIC concentrations in soil solutions, including hydrology, land cover or vegetation type, climate, temperature, terrestrial C fluxes, and soil class. In addition, a range of chemical and physical soil properties has been identified as controls of C leaching, such as soil organic carbon-to-nitrogen (C/N) ratio, soil organic carbon (SOC) content, pH, and texture (Kalbitz et al. (2000),Table 1).

In the unsaturated (or vadose) zone, dissolved C in soil pore water between the surface and the saturated groundwater zone is considered. DIC and DOC concentrations generally change with depth during transport to aquifers (Dalva and Moore, 1991; Michalzik et al., 2001; Shibata et al., 2001). DOC concentrations are highest in the upper organic soil horizon, declining by 10-50% in the subsoil (Dalva and Moore, 1991; Neff and Asner, 2001). In most soils, DOC concentrations are controlled by biodegradation in the topsoil and adsorption in the subsoil (K'O H et al., 1996; Kalbitz et al., 2000; Michalzik et al., 2001;

Sanderman et al., 2008). DIC concentrations typically increase with depth due to organc C biodegradation. Oxygen strongly impacts biogeochemical processes like decomposition in the unsaturated zone (Dalva and Moore, 1991; McLaughlin et al., 1996; Trettin and Jurgensen, 2003). Weathering can also be a source of DIC (Davidson and Trumbore, 1995; Shibata et al., 2001; Winterdahl et al., 2016).

Dissolved C fluxes from soil solutions are usually estimated using flow data for hydrology and field measurements of

concentrations (Don and Schulze, 2008). Inventories of dissolved C in soil solutions generally focus only on a specific region or environment like grasslands (Sparling et al., 2016), temperate forests (Michalzik et al., 2001; Borken et al., 2011), specific forest types (Liu and Sheu, 2003; Schrumpf et al., 2006) or other vegetation types (Neff and Asner, 2001; Aitkenhead-Peterson, 2000). These studies therefore only represent a specific environment (Chantigny, 2003; Deb and Shukla, 2011; Camino-Serrano et al., 2014). Similarly, modelling studies on dissolved C in soils focus on a specific site, region or specific environment (Grieve,

1991; Boyer et al., 1996; Currie and Aber, 1997; Michalzik et al., 2003; Wu et al., 2014; Dick et al., 2015), but a few models have been developed at the scale of large river basins, countries or parts of continents (Rowe et al., 2014; Tian et al., 2015; Stergiadi et al., 2016; Sawicka et al., 2017). Camino-Serrano et al. (2014) presented a first continent-wide data inventory and regression model for DOC in soil solutions for mainly European forests (Camino-Serrano et al., 2014; Camino Serrano et al., 2016).

However, a global data inventory and analysis of DOC and DIC in soil solutions covering a wide range of vegetation types and climates is lacking, inhibiting identifying a set of drivers explaining dissolved C concentrations at that scale (Michalzik and Matzner, 1999; Deb and Shukla, 2011; Camino-Serrano et al., 2014). The goal of this study is to estimate vadose zone dissolved C concentrations in soil solutions on the global scale. We compiled a global database on DOC and DIC concentrations and fluxes in soil solutions, representing a range of environmental conditions and including a range of potential drivers. Using

this database we developed (multi-) regression models to calculate topsoil concentrations, and concentration vs. depth in the





subsoil at the global scale. This model will be included in the Integrated Model to Assess the Global Environment (IMAGE) Dynamic Global Nutrient Model (DGNM) (Vilmin et al., in prep.).

## 2   Methods

### 2.1   Database construction

We collected published studies reporting measurements of dissolved C in soil solutions in the unsaturated zone, covering 762 entries from 351 sites distributed over the main climate zones (Figure 1 and Table 1). From these studies, all individual measurements (i.e., per location and depth) were included in the database, together with ancillary information, such as on climate, soil and land cover. Studies were selected when reporting on DOC or DIC concentrations. Additionally, some studies only reporting DOC or DIC fluxes where added. When available, measurements on related forms such as biogenic DOC or

alkalinity where also included. For the database to be representative of the spatial trends in dissolved C content, without seasonal or temporary effects (Michalzik and Matzner, 1999; Neff et al., 2000; Neff and Asner, 2001; Don and Schulze, 2008), we included only yearly average values calculated from measurements representing shorter time periods covering at least one year. A number of measurements were found not to fulfill the latter criterion and are thus reported under a different category as non-yearly averages (See Table 1). All selected data are from measurements in the unsaturated zone unless specified differently,

for example for peat soils (Histosols) with a high water table.

Entries in the database for the same site but for different depths have the same unique sampling ID. Thus, entries with the same sampling ID can be analyzed for attenuation profiles over depth. Topsoils include all entries reported as a topsoil and samples from <10 cm depth. All other soils are classified as subsoils. An alternative classification following Batjes (2016) is also included, classifying all samples measured within 20 cm as topsoils and all samples from below 20 cm as subsoils.

Information on general, environmental and soil characteristics, soil properties or data on the terrestrial C budget are included when available (see Table 2 for the available meta-data for DOC concentrations, or the database in SI 1). Based on the descriptions in the literature, we identified classifications for soil horizons, USDA soil class (Bouwman, 1990; USDA-NRCS, 2005), vegetation and, when available, soil texture (see SI 1). Several climate classifications were extracted from global climate maps representing the second half of the 20th century according to Kottek et al. (2006) (SI 1 and Figure 1).

DOC concentrations in the subsoil are calculated as relative concentrations compared to those in the topsoil. This approach allows to include data from laboratory experiments, since biases inherent to incubation experiments are probably consistent in topsoil and subsoil. Relative concentrations are calculated for locations with data for various depths, provided that the shallowest depth is <10 cm (with 10 cm taken as the median of the topsoil according to the classification following Batjes (2016)).

Soil data (both 30 second resolution and aggregations to 30 minutes, representing the mean and dominant value) from Batjes (2015) and temperature and precipitation data (New et al., 1997) (30 min, long-term average+trend; data available from the climate database of the IMAGE model (Stehfest et al., 2014)) are extracted for every database site for the corresponding grid cell. The database was compiled in Microsoft Excel, data were analyzed with codes written in R or Python.





## 2.2 Statistical analysis and model construction

All numerical variables were plotted against each other and studied using simple descriptive methods (pearson R, spearman R, mean, median, standard deviation), for all classifications. Several transformations were examined for all C data (ln, log10, square root, square, reciprocal). DOC concentration data have the largest global coverage (Table 1). Therefore, only yearly
DOC concentration data are selected for the construction of the global model . We use Box plots, QQ-plots and histograms to present and analyse frequency distributions and to identify outliers. Because most data are not normally distributed, the Mann–Whitney U test was used to compare subgroup differences. We constructed different models to calculate the annual average DOC concentrations in topsoil and subsoil.

The topsoil model is constructed in a combined forward/backward multiple-linear regression analysis, correcting with the
Akaike information criterion (AIC) for the number of factors included. Covariate non-linearity was examined using partial residual plots (PRP), taking first the variables that show the strongest non-linearity, following Fox (2015). Added variable plots (AVP) were used to identify outliers (isolated largest residuals or largest partial leverages) (Velleman and Welsch, 1981). The model performance is expressed by the coefficient of determination ($R^2$) and the root mean square error (RMSE).

The subsoil model is constructed as a relative attenuation function with depth starting from on the topsoil concentration.
Codes for data processing, regression analysis and model construction were developed in R or Python.

## 3 Results and Discussion

### 3.1 General aspects and data analysis

The database contains mainly data for yearly average DOC concentrations and fluxes (72% and 37% of the database entries), with only limited data for DIC (<11%) (Table 1). DIC data generally show concentration/flux profiles as expected, increasing
with depth due to decomposition (Davidson and Trumbore, 1995; Trumbore et al., 1995; Shibata et al., 2001; Winterdahl et al., 2016). However, the DIC data are scant, which inhibits analysis and model development on a global scale. Therefore, we selected only DOC data for further analysis. Since both the number and global coverage of DOC concentration data is much larger than that of DOC flux data, our further analysis focuses on DOC concentrations (Table 1).

Annual average DOC concentrations range from 1.7 to 88.3 (median=24.43) mg C/L for topsoils and from 0.42 to 372.1
(median=6.65) mg C/L for subsoils (excluding laboratory incubations). When excluding data from Histosols, median values are 25.27 and 5.50 mg C/L for topsoil and subsoil respectively. DOC concentrations decrease with depth from the organic rich topsoils down to the subsoils (Figure 2). Earlier, smaller dataset compilations from forests by Dalva and Moore (1991) and Michalzik et al. (2001) show similar profiles and concentration ranges. Both the whole data set and horizon sub-sets are positively skewed distributed (Figure 2 and SI 3), similar to the distribution of several soil properties in global datasets (Batjes,
2015, 2016).

A range of different techniques has been used for measuring DOC concentrations. Laboratory experiments yield significantly higher concentrations in both topsoils (p<0.001) and subsoils (p<0.001) than other methods, with five times higher median





concentrations in topsoils. This can be attributed to the disturbance and different conditions, that are often not representative for field conditions (Lawrence and David, 1996; Kane et al., 2006; Chantigny, 2003; Guggenberger and Kaiser, 2003; Jones and Willett, 2006; Chantigny et al., 2014), In a review study Kalbitz et al. (2000) conclude that relations identified in laboratory studies can often not be confirmed in the field, in particular for subsoils (Kalbitz et al., 2000). Therefore, we used data from

the laboratory experiments only for analyzing the relative concentration changes with depth.

In subsoils, piezometer-based concentrations are much higher than for other sampling methods (p<0.001, Figure 3), as these samples are almost exclusively taken in Histosols, where DOC degradation is inhibited by high groundwater tables (Easthouse et al., 1992). Where Zsolnay (2003) and Buckingham et al. (2008) found different concentrations between zero-tension and tension lysimeters (Sparling et al., 2016), we did not observe this in neither top- (p>0.1) or subsoil data (p>0.1). Thus, both

methods are combined in the analysis (Litaor, 1988a; Schrumpf et al., 2006).

No single potential driving factor showed a clear correlation with DOC concentrations in a simple (linear) regression (maximum $R^2$=0.13; see for example Figure 5). This suggests that DOC concentrations are not driven by one uniform first-order governing factor, but by a set of interrelated drivers and controls that are spatially variable (Chantigny, 2003; Ranville, 2005). For example Aitkenhead-Peterson (2000) showed for temperate forest soils that both DOC production and heterotrophic respi-

ration (HR) increase with temperature and also soil C/N ratios (Gödde et al., 1996; Aitkenhead-Peterson, 2000).

The distribution of data on potential drivers of DOC concentrations is unbalanced; the choice of factors included in sampling studies varies (Table 2). This is a problem earlier recognized on a smaller scale (Kalbitz et al., 2000; Evans et al., 2005). As a result of these data gaps, for many factors, analysis is only possible on a limited part of the DOC concentration data. Moreover, including a few factors strongly reduces the amount of data involved (e.g. for topsoils, including pH, SOC and CN cuts the

amount of data from 255 to 40), thereby impeding analysis at a global scale. Indeed, significant relations can be found for sub-sets of the data.

As we aim to constrain DOC concentrations on a global scale, we selected those factors with the largest data coverage and thus had to ignore some soil properties and environmental factors with potential predictability. Instead we use factors such as soil class, which often capture overall environmental conditions that determine soil forming factors and the resulting soil

physical and chemical characteristics (Don and Schulze, 2008; Tipping et al., 1999), and land cover/vegetation (Currie et al., 1996; Chantigny, 2003), as a proxy for the soil C cycle, and climate zones representing temperature and moisture conditions (Litaor, 1988b; Tipping et al., 1999). In other studies similar generalized controlling factors have been proposed, such as *"biological activity"* (K'O H et al., 1996), *"microbial metabolism"* (McDowell and Likens, 1988) or a *"physico-chemically dominated control"* (Michalzik and Matzner, 1999). Further, we tested the effect of temperature and precipitation. Measured

temperature and precipitation show a good correlation to corresponding climate data ($R^2$ of 0.94 and 0.49). As the measured data are not available for all measurements, we used the temperature and precipitation data from New et al. (1997) instead of the measured values in our model construction.

Concentrations of DOC in top and subsoils differ by soil class, land cover type and climate zone (Figure 4). In contrast to other soil classes, Histosols generally have higher concentrations in the subsoil compared to the topsoil (Easthouse et al., 1992;

Trettin and Jurgensen, 2003; Moore and Clarkson, 2007). This has a clear confounding effect on the data of e.g. land cover





type 'grass agriculture', which seem to have higher median concentrations in subsoils than topsoils (Figure 4). However, when we exclude the Histosols, a more consistent pattern is shown (SI 3), with concentrations clearly decreasing with depth.

## 3.2 Topsoils

### 3.2.1 Database

5 For the topsoil (alternative classification following Batjes (2016)), 255 entries on DOC concentrations are included in the database (excluding 10 laboratory incubations). The specific horizon being studied can impact the amount of DOC detected. Aitkenhead-Peterson (2000) identifies that in a range of forest floor samples, 74% of the DOC is from the organic horizon, 12 % from litter and 13 % from the deeper roots, emphasizing the role of the topsoil in DOC production. For our topsoil data, O, A and O/A horizons are dominantly present, with lowest DOC concentrations in the A horizon (Figure 6). Values for Histosols 10 do not differ significantly from the other topsoil data (p>0.1), so can be included in a further topsoil data analysis. Since B horizons are normally subsoils we excluded them from the analysis of topsoil DOC concentrations.

All main climate zones are covered in the data, although there is a bias towards temperate and continental climates (Figure 4), an issue recognized in earlier studies (Chantigny, 2003; Camino-Serrano et al., 2014). Highest concentrations occur in humid continental climates and lowest in semi-arid climates. Also, oceanic climates have higher values than the three tropical climates 15 (p<0.001). Generally, we observe higher median concentrations in more moderate climates, with some of the warmer regions having lower concentrations (Figure 4 and SI 3). This could be explained by the lower decomposition rates in temperate zones, caused by sub-optimal conditions for microbial degradation (K'O H et al., 1996; McDowell and Likens, 1988; Aitkenhead-Peterson, 2000; Litaor, 1988b). Further, a limited or absent litter layer in semi-arid, tundra, and savannah climates restricts the amount of organic C available for decomposition. Where some studies identified different DOC concentrations between 20 coniferous and deciduous forests (Currie et al., 1996; Fernández-Sanjurjo et al., 1997), this was not observed at the global scale (p>0.1), despite the large number of data entries (Figure 4), consistent with observations by Michalzik et al. (2001). We therefore aggregated the forest data into one main class (SI 1), as we can use climate zones to account for the climatic impact on the tropical and montane forest data.

### 3.2.2 Model

25 The multi-regression model was constructed involving the variables which are available for all DOC database entries. Therefore, factors like soil texture, C/N ratio or pH could not be included. Data from soil databases may be an alternative; However, soil data from Batjes (2015) (mean or dominant value for both spatial resolutions) yielded a poor correlation with observed topsoil DOC concentrations or other factors such as C/N ratio. In addition, including these extracted soil data in the multi-regression analysis yielded a poor correlation. On the basis of the data entries in the database and using the AIC to select the 30 best fitting model, four factors are included in the model for calculating DOC concentration in the topsoil soil solutions:

$$DOC_{top} \quad = \quad MAX(0.0, 1.623 + coef_{CZ} + coef_{SC} + coef_{LU} - 0.008207 P_{annual}) \qquad (1)$$



where $DOC_{top}$ is the DOC concentration in topsoil soil solutions (mg C/L), $coef_{CZ}$, $coef_{SC}$ and $coef_{LU}$ are the coëffcents for respectively main climate zones, soil class and main land cover groups (Table 3), and $P_{annual}$ is the annual average precipitation (mm/y) (New et al., 1997)). 31% of the variation is explained by the model (RMSE=14.9, RSE=15.5, 242 degrees of freedom). Neither including sub-classes for climate and land cover, nor transforming factors to non-linear functions yielded

better results.

The scatter plot of measured versus calculated DOC concentrations (Figure 7) shows that the model describes the main trend reasonable ($R^2$ = 0.36). However, the model tends to overestimate lower concentrations for temperate and continental zones, while higher concentrations (>50 mg C/L) are consistently underestimated. The main reason is that a wide range of DOC concentrations occurs within a limited range of annual precipitation. Precipitation is negatively correlated to DOC con-

centrations (p<0.001; Figure 5), though the large residuals indicate considerable uncertainty. For a $P_{annual}$ in the range of 700-1000 mm/y (mainly continental and temperate climates), DOC concentrations range from 1.7 to 87.6 mg C/L with high concentrations corresponding to the high values in continental and temperature climates shown in Figure 7. The uncertainty may be due to the absence of information on the seasonality of precipitation and the multiple ways in which precipitation may influence concentrations, e.g. temporary accumulation in dry periods and flushing in wet seasons, dilution (Kalbitz et al., 2000;

Meir et al., 2004; Don and Schulze, 2008) or via soil moisture content (Litaor, 1988b; Christ and David, 1996; Gödde et al., 1996). Another cause of uncertainty is the poor spatial coverage of some governing factors, with the result that they could not be used for spatially dependent relationships. For example, C/N ratio is an important factor in the case of Spodosols in temperate/continental climate zones ($R^2$ = 0.36), consistent with several observations (Raastad and Mulder, 1999; Aitkenhead and McDowell, 2000; Aitkenhead-Peterson, 2000; Michalzik et al., 2001; Kindler et al., 2011), although other studies found

no relationship for topsoils at the ecosystem level Michalzik et al. (2001), while others even found even negative correlations (Fröberg et al., 2006; Sawicka, 2014).

We calculated global topsoil DOC concentrations using maps on land cover (Stehfest et al., 2014), precipitation (New et al., 1997), USDA soil classes from USDA-NRCS (2005) and climate zones by Kottek et al. (2006) (Figure 8). Model results yield topsoil concentrations up to 51.1 mg C/L, with higher values generally in higher latitude continental zones with

abundant forests. Lower concentrations occur in the arid zones with limited vegetation. DOC concentrations in equatorial regions, such as rainforests, are generally below  25.0 mg C/L. In an earlier overview Camino-Serrano et al. (2014) also found DOC concentrations in tropical topsoils to be significantly lower than those in boreal or temperate topsoils (see Figure 1d Camino-Serrano et al. (2014)). This confirms that the effect of temperature on DOC is site-specific and possibly indirect (Aitkenhead and McDowell, 2000), unlike results from some early forest studies (Guggenberger and Zech, 1993; Michalzik and

Matzner, 1999). Probably, low tropical topsoil DOC concentrations are caused by optimal conditions for microbial degradation, reducing the amount of DOC before it leaches into the soil (McDowell and Likens, 1988; K'O H et al., 1996; Aitkenhead-Peterson, 2000). Topsoil concentrations in Histosols do not exceed those in mineral soils (Figure 6,8), similar to earlier findings (Camino-Serrano et al., 2014).





### 3.3 Subsoils

#### 3.3.1 Database

For the subsoil, 285 entries on annual average DOC concentrations are included in the database (including data measured in laboratory incubations). Of the reported horizons (not present for all DOC data; Figure 9), B horizons are dominant, with only

5 few data for O/A horizons present. Except for the limited amount of O/A and C horizon values, the data in the horizon classes have similar distributions and medians. Unlike for topsoils, subsoil DOC concentrations in Histosols strongly differ from other subsoils. Low biodegradation rates due to anoxic conditions, caused by high water tables, as well as the high organic matter content in Histosols result in high DOC concentrations at larger depths (Easthouse et al., 1992; Moore and Clarkson, 2007).

When excluding Histosols, higher subsoil concentrations generally relate to higher topsoil DOC concentrations (Figure

4), whereby the gradients are smaller for areas with low input due to low microbial activity, such as shrublands, tundra or permafrost (Gelisols) (K'O H et al., 1996; Aitkenhead-Peterson, 2000; Litaor, 1988b). Soils under crops are a special case, with higher DOC concentrations in subsoils than in topsoils (Figure 4). A possible explanation is DOC excretion by deeper plant roots (Undurraga et al., 2009). However, data for soils under crops are scarce and may not represent actual conditions for croplands, similar to Salazar et al. (2019).

#### 3.3.2 Model

DOC concentrations in subsoils are regulated through biodegradation and the balance of adsorption-resorption processes (Kalbitz et al., 2000; Michalzik et al., 2001; Sanderman et al., 2008). Evidence of a strong influence of specific driving factors is mainly derived from laboratory experiments and less evident than for topsoils (Kalbitz et al., 2000). As physical-chemical factors are a main control of subsoil DOC concentrations (Kalbitz et al., 2000; Neff and Asner, 2001; Sanderman et al., 2008),

soil class can therefore be used as a proxy for physical-chemical conditions (Kaiser et al., 1996). In addition, soil classes also partly account for differences in hydrological flow paths (Johnson et al., 2006). Several field studies identified differences in subsoil DOC concentrations between USDA soil classes (Easthouse et al., 1992; Tipping et al., 1999; Johnson et al., 2006; Don and Schulze, 2008).

We calculate DOC concentrations in the subsoil as relative concentrations compared to the topsoil. In other words, the rel-

25 ative concentration in the subsoil is the quotient of a concentration at depth $x$ relative to the topsoil concentration. Assuming a constant attenuation with depth within a soil class, an exponential decay function can be fitted for each soil class. Figure 10 shows the example for Spodosols, with the corresponding coefficients for all classes in Table 4. In Histosols, DOC concentrations in subsoils can be both smaller or larger than those in topsoils (Table 4). With a known topsoil concentration for a corresponding soil class, the DOC subsoil concentration at depth $x$ is calculated as follows:

$$DOC_{sub} = DOC_{top}\, e^{(x\, coef_{SCsub})} \tag{2}$$

where $DOC_{sub}$ is the DOC concentration at depth $x$ in the unsaturated zone, $DOC_{top}$ is the topsoil concentration as measured or modelled by equation 1, and $coef_{SCsub}$ is the soil class-dependent decay coefficient (Table 4).





For mineral soils, $R^2$ values of the regressions range from 0.23 (Oxisols) to 0.74 (Alfisols and Spodols), up to 0.96 (Vertisols, but limited data) (Table 4). For some soil classes, in particular Vertisols, Gelisols and Andisols, the analysis is based on a limited dataset (Table 4) reflecting the small area covered by these soils (Vertisols 2%, Andisols 1%) or the permanently frozen state of the soil (Gelisols) (Bouwman, 1990; USDA-NRCS, 2005). Except for Histosols and Oxisols, the functions describe patterns

of the relative concentrations vs. depth quite well with no consistent overestimation nor underestimation for undeep soils. For deeper subsoils (>1.0m) the model seems to consistently underestimate DOC concentrations, for example due to DOC production or exudation by plant roots (Undurraga et al., 2009) or release due to desorption from soil colloids (Koprivnjak and Moore, 1992; Nelson et al., 1992; Neff and Asner, 2001; Sparling et al., 2016). In addition, the degradation of DOC could be overestimated by ignoring increasing recalcitrance of the OC with depth (Sanderman et al., 2008; Catalán et al., 2016).

Data and profiles for Oxisols and Histosols clearly differ from the other soil classes (SI 5) because in the strongly weathered Oxisols relatively high concentrations occur at greater depths due to high water percolation rates (Johnson et al., 2006). In Histosols, relative concentrations are observed to be both above (72% of the data) and below (28%) 1.0, with a very low $R^2$ (0.01), depending strongly on the depth of the water table (see footnote 3 in Table 4).

Using equation 2, the modelled topsoil DOC concentrations (Figure 8), k-values (Table 4) for USDA soil classes from

15 USDA-NRCS (2005) we calculated the subsoil DOC concentrations at a depth of 1 m (Figure 11). DOC concentrations at this depth range up to 265.5 mg C/L, though values above 20 m/L only occur in areas with abundant Histosols. Highest subsoil concentrations in non-Histosols are between 5 and 10 mg C/L and mainly found in permafrost areas (Figure 8).

The high concentrations in subsoils of Histosols are consistent with observations by Camino-Serrano et al. (2014), who attributed the high DOC concentrations in subsoils to the low hydraulic conductivity in these soils. Histosol concentrations vs.

20 depth of dissolved compounds with depth strongly depend on the water table depth (Trettin and Jurgensen, 2003) and therefore the thickness of the oxic zone, and can thus both increase or decrease with depth (Easthouse et al., 1992; Moore and Clarkson, 2007). High concentrations in subsoil Gelisols are similar to observations of Stutter and Billett (2003) and MacLean et al. (1999) and caused by reduced DOC degradation due to low microbial activity in the permafrost (Petrone, 2005).

Our subsoil model uses soil classes, which inherently include effects of climate and vegetation. For example, Oxisols and

25 Ultisols are typical subtropical and tropical soils (Johnson et al., 2006), while Aridisols are typical for (semi)deserts with generally low vegetation coverage (USDA-NRCS, 2005; Stehfest et al., 2014).

### 3.4 Application and perspective

Our global database on dissolved C in soil solutions does not yet include some other factors that have been identified in the literature as potential controls of DOC. For example, cation exchange capacity (CEC) (Kahle et al., 2004), terrestrial acid

deposition (Sawicka et al., 2016), anion deficit (Marin et al., 1990; Fujii et al., 2008) , soil specific surface area (Nelson et al., 1992) or the DOC composition (McLaughlin et al., 1996; Fellman et al., 2008). Still, only few studies included in our database include a broad range of parameters, measured in the similar way. To enable further in-depth analysis and include additional process controls on a global scale, a standardized set of ancillary data and uniform sampling method is required. The ICP Forests program set up such a framework for monitoring in European forests (Nieminen et al., 2016), which enabled recent


in-depth analysis and model construction on this sub-continental scale (e.g., Camino-Serrano et al., 2014; Camino Serrano et al., 2016; Sawicka et al., 2016, 2017; Johnson et al., 2018).

Our model simulates DOC concentrations at the global scale, which is a major step forward compared to current, recent large-scale (country, large basin, sub-continental region) models (e.g. Rowe et al., 2014; Tian et al., 2015; Stergiadi et al.,

2016; Sawicka et al., 2017). The temporal scale is one year. Temporal downscaling of the yearly average modelled DOC concentrations could be based on relative seasonal variability (e.g. Sawicka et al., 2016). When hydrology (e.g. Van Beek et al., 2011; de Graaf et al., 2017) is combined with our model, we could model the global DOC fluxes and also constrain the C fluxes from the terrestrial to the aquatic system.

## 4 Conclusions

We present the first global database on annual average DOC and DIC in soil solutions, covering all main climate zones. As data on DIC are scant, we conducted our analysis and model construction on annual average DOC concentrations. Highest topsoil DOC concentrations occur in forests in humid continental climates, while topsoils of Histosols do not have higher DOC concentrations than other soil classes. In contrast, highest concentrations in subsoils occur in Histosols. Our analysis shows that DOC concentrations are controlled by a complex of processes that vary in space. DOC concentrations from laboratory

experiments are consistently higher than values found in the field.

We identified a set of four distant controls of global topsoil DOC concentrations, i.e. precipitation, climate zones, vegetation types and soil classes. Further, our analysis showed that global subsoil DOC concentrations vs. depth can be modelled for all USDA soil classes. Here, soil class represents generalized physico-chemical properties that are not represented when only climate or land cover are used. Future sampling studies on DOC should be conducted in regions with land cover types cur-

20 rently underrepresented, such as crops, preferably over different soil classes. A standardized set of ancillary data and uniform sampling method would enable further constraining of global dissolved C concentrations in soil solutions.

*Supplementary material and data availability.* Supplementary material, including the database, is is being made publicly available on PAN-GAEA (https://www.pangaea.de/).

*Author contributions.* JL, AFB, AHWB and JJM designed the study. Database compilation was conducted by JL. Code writing and data

extraction from grid was done by JL, AHWB, JMM and LV. All authors contributed to the analysis and discussion of the results. JL wrote the manuscript. AFB, LV, JJM contributed to the revision of the manuscript.

*Competing interests.* The authors declare no competing interests.



*Acknowledgements.* This research was funded by the NWO New Delta 2014 project no. 869.15.015. W.J.H was funded by the NWO New Delta 2014 project no. 869.15.014. A.F.B. and A.H.W.B. received support from the PBL Netherlands Environmental Assessment Agency through in-kind contributions to The New Delta 2014 ALW projects no. 869.15.015 and 869.15.014. L.V. was funded by the Earth and life sciences (ALW) Open Programme 2016 project no. ALWOP.230 We are grateful to Marta Camino-Serrano (Centre for Ecological Research and Forestry Applications, CREAF) for providing us with a list of several studies to include in the database, which we acknowledge her by mentioning it in the database references concerned. We further thank Peter Janssen (PBL) and Maarten Zeylmans Van Emmichoven (Physical Geography - UU) for assisting in the analysis design and data extraction from maps respectively.





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





**Figures**

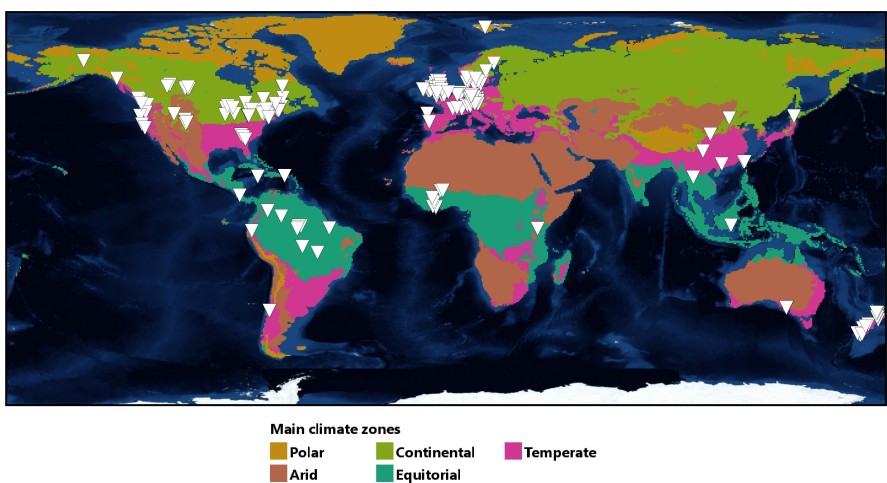

**Figure 1.** Sites reported in the database, distributed over main Köppen-Geiger climate zones (Kottek et al., 2006).





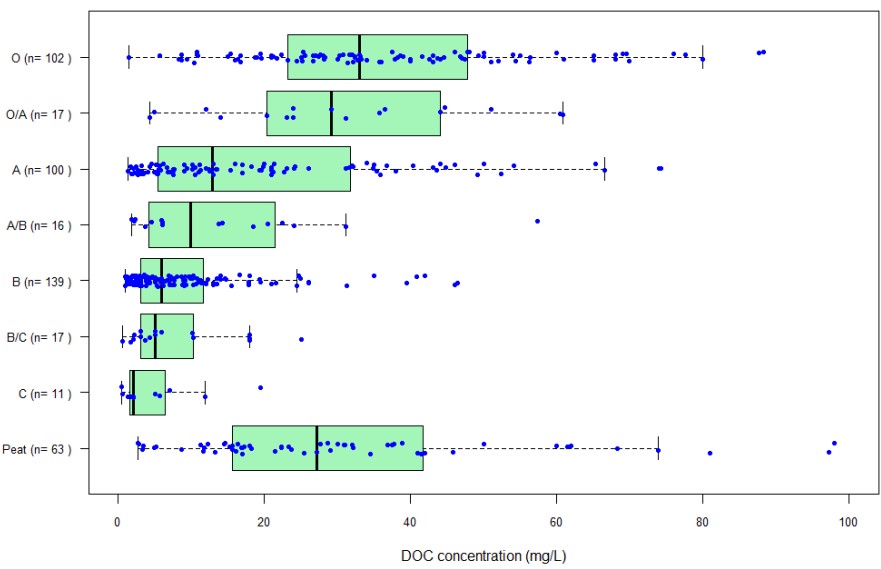

**Figure 2.** DOC concentration (mg C/L) data distributed over horizon classes (n=465, 85% of all DOC concentration data). Boxplot bars are medians. Whiskers extend up to 1.5 times the interquartile range (IQR) from the lower and upper quartile, unless exceeding the minimum or maximum. Individual data are jittered for visualization purposes. Five high non-lab-incubation values from peat samples (up to 372 mg C/L) are outside the figure range, but included in the distribution.



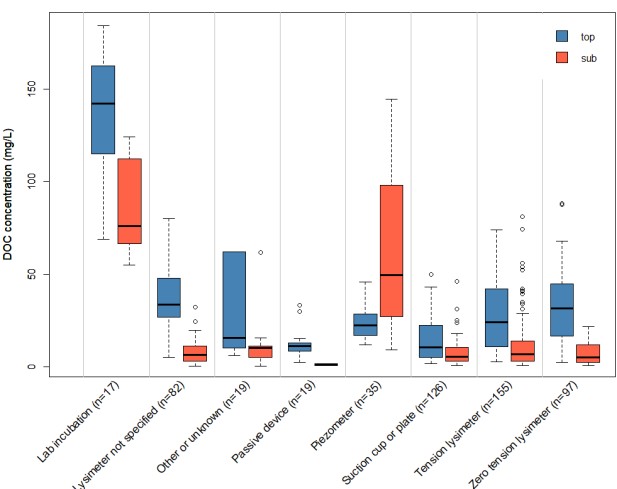

**Figure 3.** Top and subsoil DOC concentration (mg C/L) data distributed per measuring method (n=550). Boxplot bars are medians. Whiskers extend up to 1.5 times the interquartile range (IQR) from the lower and upper quartile, unless exceeding the minimum or maximum. Circles are values exceeding the whiskers. Two subsoil values (up to 372 mg C/L) from Histosols are not shown, but included in the distribution. 'Passive device' is a shorter name for the group 'Passive bottle/well/tray/ditch/gauge/plate'.

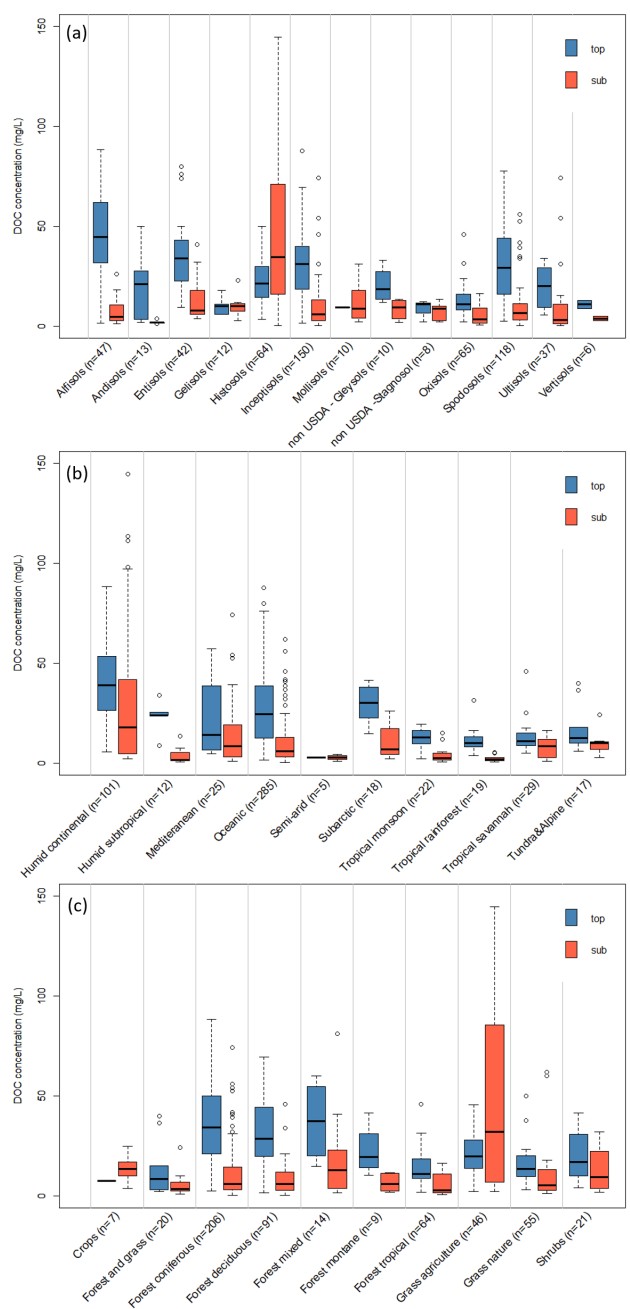

**Figure 4.** Top and subsoil DOC concentration (mg C/L) data distributed over (a) USDA soil classes, (b) sub-climate zones according to Kottek et al. (2006) and (c) land cover groups. Laboratory incubation data are excluded, n=533. Boxplot bars are medians. Whiskers extend up to 1.5 times the interquartile range (IQR) from the lower and upper quartile, unless exceeding the minimum or maximum. Circles are values exceeding the whiskers. Two subsoil values (up to 372 mg C/L) from Histosols are not shown, but included in the distribution. In a), data for soils with a double or mixed classification is represented in both single classes. See SI 3 for similar distributions as for b) and c), but excluding Histosols.





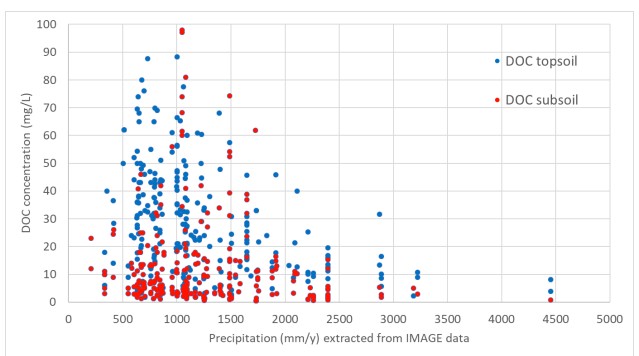

**Figure 5.** Precipitation data from New et al. (1997) vs DOC concentrations (mg C/L) (n=533). Laboratory incubation data are excluded. Five DOC subsoil peat values (111.25 - 372.1) are not shown, but included in the regression. $R^2$ values for all data, topsoil data and subsoil data are respectively 0.0325, 0.129 and 0.0073, only significant for all data and topsoil data (p<0.0001)

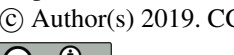


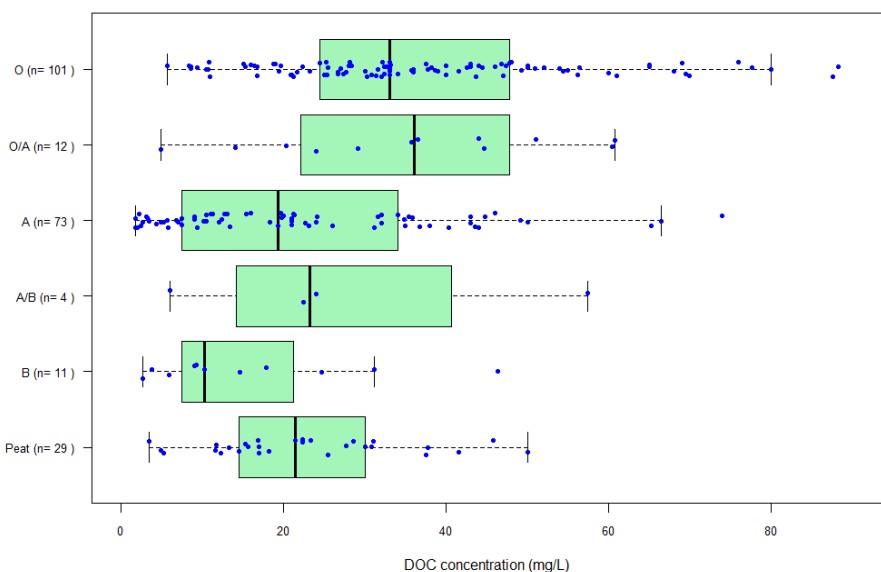

**Figure 6.** Topsoil DOC concentration (mg C/L) data distributed over horizon classes (n=230, 90% of all topsoil DOC concentration data). Laboratory incubation data are excluded. Boxplot bars are medians. Whiskers extend up to 1.5 times the interquartile range (IQR) from the lower and upper quartile, unless exceeding the minimum or maximum. Individual data are jittered for visualization purposes.



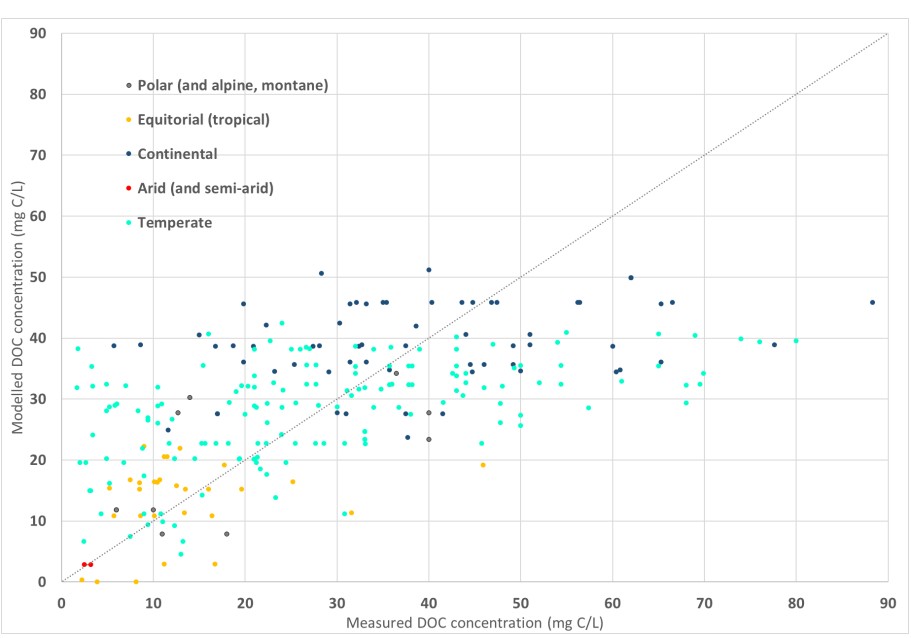

**Figure 7.** Modelled topsoil DOC concentration (mg C/L) vs observed values (n=264), distributed over the five main climate zones (Kottek et al., 2006). $R^2$ = 0.36, dotted line is 1:1 line. Entries with reported double/mixed USDA soil classes (n=34) are duplicated to enable including soil classes in the regression analysis. Laboratory incubation and B horizon data are excluded (see text). Entry #220 is also excluded as identified as a high leverage point in a partial regression analysis using an added variable plot (AVP).

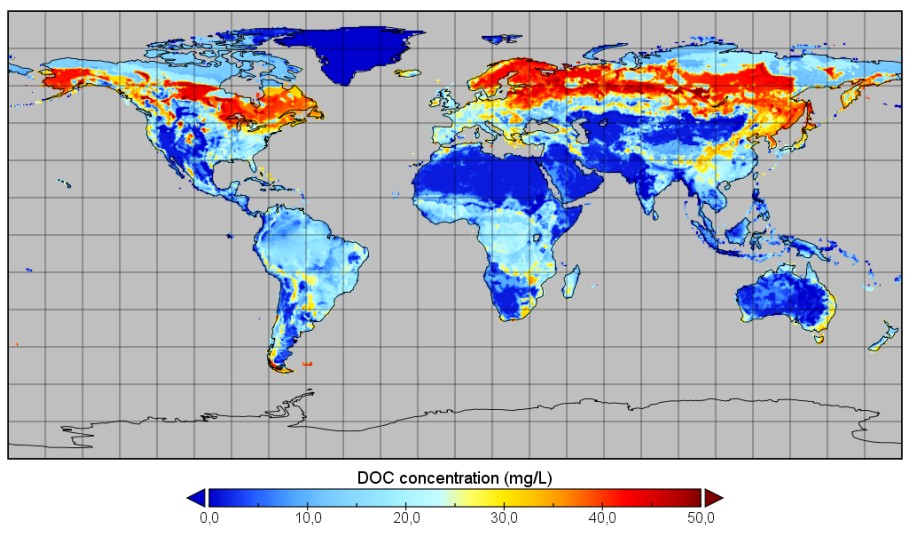

**Figure 8.** Modelled global topsoil DOC concentrations in soil solutions (mg C/L) for the year 2000 with 0.5 by 0.5 degree resolution.



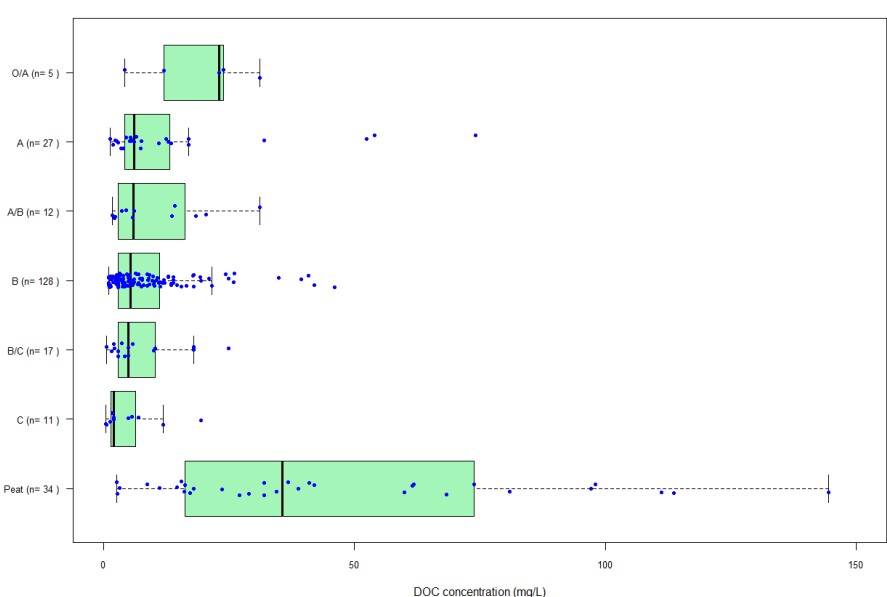

**Figure 9.** Subsoil DOC concentration (mg C/L) data distributed over horizon classes (n=234, 82% of all subsoil DOC concentration data). Laboratory incubation data are excluded (n=7). Values for classes 'O'(n=1) and 'groundwater'(n=1). Two high peat values (372.1 and 222.5 mg C/L) are not shown but included in the distribution. Boxplot bars are medians. Whiskers extend up to 1.5 times the interquartile range (IQR) from the lower and upper quartile, unless exceeding the minimum or maximum. Individual data are jittered for visualization purposes.





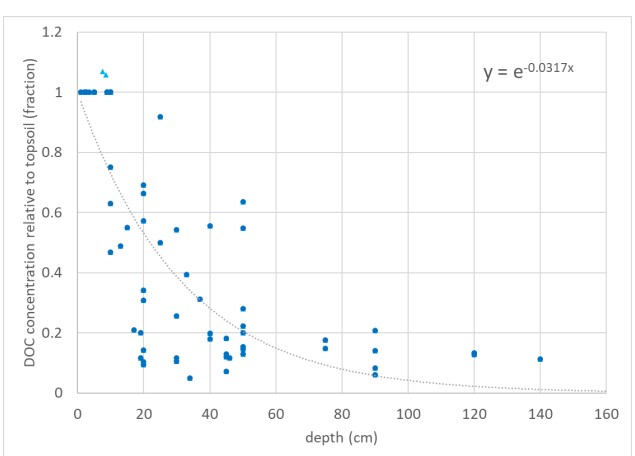

**Figure 10.** Data and model for DOC concentrations with depth, relative to topsoil concentrations. Example for Spodosol soil class data (n=81). $R^2 = 0.74$ (exponential regression). Two values above 1 (triangles) are excluded from the regression.



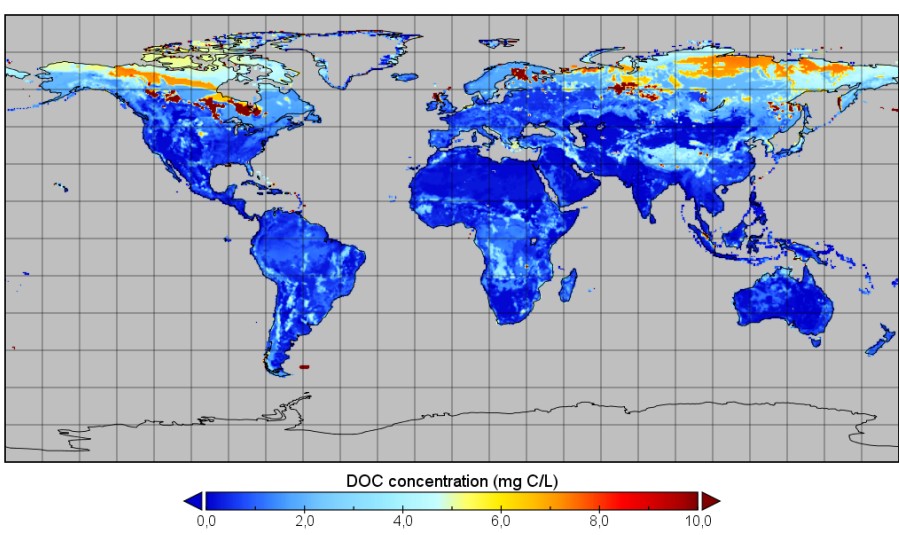

**Figure 11.** Modelled global subsoil DOC concentrations in soil solutions (mg C/L) at a depth of one meter. Based on equation 2 using topsoil data from Figure 8.





**Tables**

**Table 1.** Number of entries and sites the in database for all carbon leaching variables, and their coverage over all main climate zones.

| Dissolved carbon variable | Number of samples/entries | Number of sites | Number of main climate zones covered |
|---|---|---|---|
| DOC concentration (mg L$^{-1}$) | 550 | 229 | 5 |
| DOC flux( g C m$^{-2}$ y$^{-1}$) | 280 | 126 | 3 (continental, temperate, equatorial) |
| (total) DIC concentration (mg L$^{-1}$) | 40 | 18 | 3 (continental, temperate, equatorial) |
| (total) DIC flux (g C m$^{-2}$ y$^{-1}$) | 29 | 14 | 2 (continental and temperate) |
| biogenic DIC concentration (mg L$^{-1}$) | 4 | 4 | 1 (equatorial) |
| biogenic DIC flux (g C m$^{-2}$ y$^{-1}$) | 11 | 11 | 1 (temperate) |
| alkalinity concentration (mg L$^{-1}$) | 82 | 33 | 4 (arid, temperate, equatorial, polar) |
| alkalinity flux (g C m$^{-2}$ y$^{-1}$) | 2 | 2 | 1 (equatorial) |
| DOC concentration - not yearly avg (mg L$^{-1}$) | 93 | 51 | 5 |
| DOC flux - not yearly avg (g C m$^{-2}$ period$^{-1}$) | 39 | 28 | 2 (continental and temperate) |





Table 2: Meta-data overview, database selection of yearly DOC concentrations (550/762 entries). $N_e$=number of entries; $N_s$=number of studies.

| Factor or variable reported | Topsoil $N_e$ | Topsoil $N_s$ | Subsoil $N_e$ | Subsoil $N_s$ | All $N_e$ | All $N_s$ | Studies that identify this factor as a main control of soil solutions DOC concentration[10] | Studies that identify this factor as a secondary or unimportant control of soil solutions DOC concentration[10] |
|---|---|---|---|---|---|---|---|---|
| **General** | | | | | | | | |
| Entry ID | 265 | 188 | 285 | 193 | 550 | 229 | | |
| Reference | 265 | 188 | 285 | 193 | 550 | 229 | | |
| Location ID | 265 | 188 | 285 | 193 | 550 | 229 | | |
| Sampling location | 265 | 188 | 285 | 193 | 550 | 229 | | |
| Coordinates (reported or looked up)[1] | 265 | 188 | 285 | 193 | 550 | 229 | | |
| First sampling year | 265 | 188 | 285 | 193 | 550 | 229 | | |
| Last sampling year | 265 | 188 | 285 | 193 | 550 | 229 | | |
| Sampling period / incubation time for lab studies | 219 | 153 | 249 | 162 | 468 | 192 | | |
| Reported sampling method | 256 | 179 | 277 | 185 | 533 | 218 | No driver[11] | No driver[11] |
| Sampling method classification[2] | 256 | 179 | 277 | 185 | 533 | 218 | No driver[11] | No driver[11] |
| Measuring frequency | 170 | 111 | 191 | 133 | 361 | 146 | | |
| Samples with more entries for this location[3] | 265 | 188 | 285 | 193 | 550 | 229 | | |
| **Environmental properties** | | | | | | | | |
| Temperature (°C)[4] | 202 | 144 | 228 | 154 | 430 | 175 | (Guggenberger, 1992; Guggenberger and Zech, 1993; Cronan, 1985; Michalzik and Matzner, 1999; Michalzik et al., 2001; Sawicka, 2014) | (Aitkenhead-Peterson, 2000) |
| Water drainage flux (mm/y) | 27 | 19 | 65 | 48 | 92 | 49 | (McDowell and Likens, 1988; Currie and Aber, 1997; Michalzik et al., 2001) | (Michalzik and Matzner, 1999; Don and Schulze, 2008) |
| Precipitation (mm/y)[4] | 191 | 139 | 221 | 142 | 412 | 164 | (Sawicka, 2014) | |
| Reported climate zone | 76 | 59 | 93 | 60 | 169 | 69 | (Litaor, 1988b; Tipping et al., 1999) | |
| Climate zone classification[2] | 265 | 188 | 285 | 193 | 550 | 229 | (Litaor, 1988b; Tipping et al., 1999) | |
| Reported biome/vegetation | 265 | 188 | 285 | 193 | 550 | 229 | (Currie et al., 1996; Aber et al., 1989; Fernández-Sanjurjo et al., 1997; McDowell et al., 1998; Chantigny, 2003) | (Michalzik et al., 2001; Sawicka, 2014; Salazar et al., 2019) |
| Land cover classification[2] | 265 | 188 | 285 | 193 | 550 | 229 | (Currie et al., 1996; Aber et al., 1989; Fernández-Sanjurjo et al., 1997; McDowell et al., 1998; Chantigny, 2003) | (Michalzik et al., 2001; Sawicka, 2014; Salazar et al., 2019) |
| **Soil properties** | | | | | | | | |
| Topsoil or subsoil (old definition) | 265 | 188 | 285 | 193 | 550 | 229 | See 'Sampling depth' | |
| Topsoil or subsoil (updated definition) | 265 | 188 | 285 | 193 | 550 | 229 | See 'Sampling depth' | |
| Reported soil type (class) | 265 | 188 | 285 | 193 | 550 | 229 | (Easthouse et al., 1992; Tipping et al., 1999; Johnson et al., 2006; Don and Schulze, 2008) | (Moore et al., 2008) |





Table 2 continued: Meta-data overview, database selection of yearly DOC concentrations (550/762 entries). $N_e$=number of entries; $N_s$=number of studies.

| Factor or variable reported | Topsoil | | Subsoil | | All | | Studies that identify this factor as a main control of soil solutions DOC concentration[10] | Studies that identify this factor as a secondary or unimportant control of soil solutions DOC concentration[10] |
|---|---|---|---|---|---|---|---|---|
| | $N_e$ | $N_s$ | $N_e$ | $N_s$ | $N_e$ | $N_s$ | | |
| USDA soil classification[2] | 265 | 188 | 285 | 193 | 550 | 229 | (Easthouse et al., 1992; Tipping et al., 1999; Strobel et al., 2001; Johnson et al., 2006; Don and Schulze, 2008) | |
| Other soil properties (texture) | 147 | 104 | 166 | 109 | 313 | 123 | (Don and Schulze, 2008) | |
| Soil texture classification[2] | 141 | 98 | 162 | 105 | 303 | 117 | (Don and Schulze, 2008) | |
| Reported horizon | 224 | 160 | 214 | 151 | 438 | 188 | See 'Sampling depth' | |
| Horizon classification[2] | 236 | 163 | 236 | 168 | 472 | 204 | See 'Sampling depth' | |
| Sampling depth (cm)[5] | 249 | 172 | 267 | 180 | 516 | 206 | (Dalva and Moore, 1991; K'O H et al., 1996; Kalbitz et al., 2000; Michalzik et al., 2001; Neff and Asner, 2001; Sanderman and Amundson, 2008) | |
| C/N ratio soil[6] | 92 | 67 | 68 | 49 | 160 | 79 | (Raastad and Mulder, 1999; Aitkenhead and McDowell, 2000; Aitkenhead-Peterson, 2000; Michalzik et al., 2001; Kindler et al., 2011) (positive) (Fröberg et al., 2006; Sawicka, 2014) (negative) | (Michalzik et al., 2001; Moore et al., 2008; Kindler et al., 2011) |
| Soil organic carbon content (%)[7] | 67 | 55 | 89 | 60 | 156 | 71 | (Moore et al., 1992; Neff and Asner, 2001; Ranville, 2005; Liu et al., 2013) | (Michalzik et al., 2001; Kindler et al., 2011; Sawicka, 2014) |
| pH soil (occasionally in solution)[8] | 227 | 160 | 202 | 143 | 429 | 185 | (Brooks et al., 1999; Lofts et al., 2001; Michalzik et al., 2001; Lu et al., 2013) (positive) (Cronan, 1985; Antweiler and Drever, 1983; Marin et al., 1990) (negative) | (Cronan, 1985; McDowell and Likens, 1988; Michalzik and Matzner, 1999) |
| Fe, Al in soil or soil solutions | 83 | 65 | 69 | 59 | 152 | 72 | (Grieve, 1990; Koprivnjak and Moore, 1992; Moore et al., 1992; Kaiser et al., 1996; Neff and Asner, 2001; Major et al., 2010; Kindler et al., 2011; Lu et al., 2013) | (David and Driscoll, 1984; Dalva and Moore, 1991; Lu et al., 2013) |
| Fe, Al in soil solutions (mg C/L) | 27 | 26 | 23 | 21 | 50 | 26 | | |
| Fe, Al in soil (g/kg) | 46 | 29 | 30 | 22 | 76 | 29 | | |
| Fe, Al in soil (%) | 10 | 10 | 10 | 10 | 20 | 11 | | |
| *Terrestrial C budget elements* | | | | | | | | |
| Heterotrophic respiration (gCm$^{-2}$*y$^{-1}$) | 2 | 2 | 0 | 0 | 2 | 2 | (Currie and Aber, 1997; Sato and Seto, 1999; Aitkenhead-Peterson, 2000; Kang et al., 2001; Neff and Asner, 2001) | |





Table 2 continued: Meta-data overview, database selection of yearly DOC concentrations (550/762 entries). $N_e$=number of entries; $N_s$=number of studies.

| Factor or variable reported | Topsoil | | Subsoil | | All | | Studies that identify this factor as a main control of soil solutions DOC concentration[10] | Studies that identify this factor as a secondary or unimportant control of soil solutions DOC concentration[10] |
|---|---|---|---|---|---|---|---|---|
| | $N_e$ | $N_s$ | $N_e$ | $N_s$ | $N_e$ | $N_s$ | | |
| NPP (gCm$^{-2}$*y$^{-1}$)[9] | 2 | 1 | 3 | 1 | 5 | 1 | | |
| NEP (gCm$^{-2}$*y$^{-1}$)[9] | 2 | 1 | 3 | 1 | 5 | 1 | | |
| NEE (gCm$^{-2}$*y$^{-1}$)[9] | 6 | 6 | 17 | 12 | 23 | 12 | | |
| NBP or NECB (gCm$^{-2}$*y$^{-1}$)[9] | 6 | 6 | 18 | 13 | 24 | 13 | | |
| *Dissolved C (selected only yearly avg concentrations)* | | | | | | | | |
| Total DIC concentration (mg C/L) | 13 | 11 | 27 | 18 | 40 | 18 | | |
| DOC concentration (mg C/L) | 265 | 188 | 285 | 193 | 550 | 229 | | |
| Biogenic DIC concentration (mg C/L) | 2 | 2 | 2 | 2 | 4 | 2 | | |
| Alkalinity concentration (mg C/L) | 35 | 30 | 45 | 30 | 80 | 32 | | |

[1] Location coordinates as reported in the study, or obtained from using the location description in the study.

[2] See classification tables in SI

[3] Where measured at one site at different depths, every observations is a single entry with the same sampling ID.

[4] Measured on the site or from weather station data.

[5] Sampling depth is reported when available. Shallow or surface level observations with actual depth not reported, while sampling depths are given for other soil layers, are assumed to be 2.5 cm.

[6] For 3 entries C/N ratio in soil solutions is used instead

[7] Studies report SOC using three different units. When converting from kg/m2 to % we assumed a layer thickness of 10 cm and a fixed bulk density of 1.0 (topsoil) and 1.3 (subsoil), except for Andisols or Histosols (0.7 kg/dm$^3$).

[8] Soil pH measured in wet soil. Some measurements in CaCl$_2$ where transformed using formula 1 by Ahern et al. (1995). A number of studies reports pH measured in solution instead of in soil (14% all pH data). As these are strongly related (Easthouse et al., 1992; Lu et al., 2013), they are included when no soil pH is available.

[9] NPP=Net Primary Production; NEP=Net Ecosystem Production; NEE=Net Ecosystem Exchange; NBP or NECB=Net Biome Production or Net Ecosystem Carbon Balance, without leaching.

[10] Mentioned studies sometimes consider relations in only top- or a subsoil.

[11] Though not a driver, several studies discuss the impact of the sampling method.

See also SI 1.



**Table 3.** Regression coefficients for the topsoil model in equation 1

| USDA soil classes | $coef_{SC}$ | Climate zones [1] | $coef_{CZ}$ | Land use class | $coef_{LU}$ |
|---|---|---|---|---|---|
| Alfisols | 0.00 | Arid (and semiarid) | 0.00 | Crops | 0.00 |
| Andisols | -15.27 | Continental | 27.90 | Forest | 24.55 |
| Entisols | -2.07 | Equitorial (tropical) | 16.67 | Forest grass shrubs | 16.14 |
| Gelisols | -32.61 | Polar (and alpine, montane) | 29.36 | Grass agriculture | 25.08 |
| Histosols | -11.52 | Temperate | 21.03 | Grass nature | 12.20 |
| Inceptisols | -9.54 | | | | |
| Mollisols | -19.21 | | | | |
| Oxisols | -7.94 | | | | |
| Spodosols | -6.44 | | | | |
| Ultisols | -8.25 | | | | |
| Vertisols | -25.76 | | | | |
| Aridisols | n.a.[2] | | | | |
| | | | | | |
| *Other*[3] | | | | | |
| non USDA - Gleysols | -14.28 | | | | |
| non USDA -Stagnosol | -29.41 | | | | |

[1] Main Köppen-Geiger climate zones (Kottek et al., 2006).

[2] Aridisols are not included in the database. As these soils occur in arid to semi-arid regions with sparse vegetation and very low microbial activity, we assume the DOC concentration to be equal to that in precipitation. For the concentration in precipitation, we calculated the median value from the overview by Aitkenhead-Peterson et al. (2003); 1.55 mg C/L. For comparison, this is up to 50% lower than the database DOC values, available for a steppe climate, but in oxisols. A fixed value of 1.55 mg C/L is also used for ice cover (land use or soil class maps), rock cover, shifting sands, salt plains (soil class maps) or hot deserts (land use maps).

[3] For some DOC data, only classification based on groundwater behavior (non-USDA) was possible.





**Table 4.** Regression coefficients for the subsoil model in equation 2, based on the nonlinear (weighted) least-squares estimates.

| USDA soil class | k-value | n[1] | RMSE[1] | R$^2$ |
|---|---|---|---|---|
| All | -0.0267 | 368 | 0.39 | 0.35 |
| Average: all except obvious Histosols[2] | -0.0276 | 355 | 0.32 | 0.45 |
| Alfisols | -0.0436 | 34 | 0.26 | 0.74 |
| Andisols | -0.0273 | 17 | 0.27 | 0.51 |
| Entisols | -0.0186 | 42 | 0.25 | 0.58 |
| Gelisols | -0.0098 | 10 | 0.14 | 0.43 |
| Histosols[3] | 0.0059 | 26 | 0.74 | 0.01 |
| Inceptisols | -0.0298 | 115 | 0.37 | 0.41 |
| non USDA - Gleysols | -0.0230 | 8 | 0.30 | 0.51 |
| Spodosols | -0.0317 | 81 | 0.21 | 0.74 |
| Ultisols | -0.0353 | 19 | 0.16 | 0.85 |
| Vertisols | -0.0111 | 6 | 0.08 | 0.96 |
| Oxisols | -0.0275 | 52 | 0.43 | 0.23 |
| Mollisols - average value[4] | -0.0276 | n.d. | n.d. | n.d. |
| other - average value[5] | -0.0436 | n.d. | n.d. | n.d. |

[1] n= amount of database values of this class for which a relative reduction can be calculated. RMSE = root-mean-square error, measure of the differences between values predicted and observed. $p<0.01$, except for Gelisols ($<0.1$) and Histosols (see [3]).

[2] Average coefficient of regression on the data, excluding values in peat. Mixed soil classes including a Histosol are included.

[3] Out of all subsoil Histosol data for which a relative quotient with depth can be calculated, 28% has a relative reduction below 1.0. We therefore conducted the regression over all data, resulting in a very low positive coefficient, causing a quasi-linear positive relation. The p value is quite high (0.11) due to the large spread of both positive and negative data.

[4] Insufficient subsoil data in Mollisols is available in the database; the average coefficient without Histosols is therefore used. n.d. means no data in the database available.

[5] the USDA soil class map contains also the classes salt, shifting sand, rock and ice. Rock and ice are assumed to have no subsoil (NaN). For shifting sand and salt, the average coefficient without Histosols is used. n.d. means no data in the database available.