# Peer review of "Global database and model on dissolved carbon in soil solution"

_Biogeosciences, 2019_

## Short Comment (SC1) · 1 Jul 2019

It is a very interesting study and fits well to the scope of BG. I am not an expert on soil data and thus waiting for the comments from reviewers on the validity and robustness of the method. I have one suggestion from the modeling perspective. To date, many Earth system and land surface models are using SOC as a proxy to calculate the DOC concentration in the soil and then using it with water flow to calculate DOC fluxes to inland waters. Could the authors calculate the correlation between the modeled global DOC in this study and global SOC data (such as HWSD SOC) to see how the previous proxy-based method might bias DOC flux estimates?

Zeli Tan (zeli.tan@pnnl.gov) Pacific Northwest National Laboratory

---

## Referee Comment (RC1) · Anonymous Referee #1 · 15 Jul 2019

The authors have done a lot of work to assemble the data. The database will be a valuable resource. It is disappointing that the flux data are not analysed, since these are arguably more important than concentrations in the context of the carbon cycle. It would be of great interest to know the distribution and statistics of the flux data. I do not think the justification for analysing [DOC] only (page 4, line 22) is convincing. The authors say that fluxes can be obtained as the products of concentration and water flux. What happens if you do this, how do the results compare with the direct flux data? Looking at Figure 5, average fluxes might be > 20 gC/m2/a, which is about twice the value that Buckingham et al reported for non-peat UK soils. Buckingham et al compared straight average and flux-weighted concentrations, and found not much difference, but this was the UK only. Is it possible for the authors to make such a

comparison for their collated data? A discussion of the possible dangers of reporting straight-average annual [DOC], which I assume is what has been done, would be helpful. The model is unimpressive – the predictions do not vary much and so there is strong bias, predictions being too high at low [DOC] and too low at high [DOC]. A problem might arise if high soil moisture promotes DOC production (positive effect) while associated high water throughput causes dilution (negative effect). Again, it would be of interest to know the implications of the modelled data for flux estimation. Can the different averages in Figure 4 be compared for statistical significance? Page 4, line 20. I do not understand this argument – surely there is more decomposition of organic matter near to the soil surface, not with increasing depth? Is it more to do with escape of CO2 as gas, and the dissolved DIC flux, becoming less efficient with depth? The paper is not that well written, it could do with a careful edit for language and grammar. The word "equatorial" is mis-spelled as "equitorial" throughout. Reference S. Buckingham, E. Tipping, J. Hamilton-Taylor Concentrations and fluxes of dissolved organic carbon in UK topsoils. Sci Tot Environ 407 (2008) 460 – 470

―――――――――――――――――――

---

## Referee Comment (RC2) · Anonymous Referee #2 · 20 Jul 2019

The authors present a database which they compiled for soil dissolved organic and inorganic carbon alongside a regression model to calculate top and sub soil DOC concentration. I find the paper poorly written and organised and the model is not well constructed, missing major parameters controlling soil DOC. While I appreciate the effort of data collection which could be useful for model evaluation, I do not find model outputs and analyses reliable and do not recommend publishing this manuscript in Biogeosciences in the present format.

Major comments:

The main problems that I see with this manuscript are as follows: Authors constructed a model based on oversimplified parameters which are not fully representative of the processes that are controlling soil DOC. Additionally, they did not use a proper climate

data period. Perhaps their model could work in theory, but in this case it has not as can be seen in results. The bright side of this work was the time that authors spent on collecting the measurements from various studies, although most of them were already reported by Camino-Serrano et al. (2014).

Following some major comments:

1) I search the author's name and paper name on PANGAEA.de and I could not find any database or model codes. It will be useful if the authors provide a direct link to such data. However, I extracted the model results from the provided netcdf file attached to this manuscript. First of all, the time dimension in the model output has 7 layers which seem almost identical to me. What are those 7 layers and why are they almost identical?. Secondly, the model output is for 1970 to 2000 while their database is not in this time period. Thirdly, the model outputs are significantly lower than their reported database and previous studies. I get average of 16.5 mg C L-1 (median:13.7) for the topsoil and average of 2.1 mg C L-1 (median: 0.9) at the subsoil layer. This shows the weakness of model in representing the processes which are controlling the DOC concentration (mentioned above).

2) I noticed in your dataset there are many sites that you only have DOC flux reported. If you do not want to analyse this flux, all those sites should be removed from your dataset. Keeping just sites with observed DOC concentrations will leave only 550 measured points where plenty of them have same sample ID. Hence the claim of having 762 entries and 351 sites is not really true for soil DOC. Also 94 points are non-yearly averaged, with at least 40 of them are only once measured. These data should be removed from data analysis as they cannot be representative of a site. The future corrected model should be run again excluding these points and all result should be corrected.

3) I do not see any model vs measurements validation for subsoil.

Comments/Question:

p1.l15: 2.9 Pg C yr-1 is not the processed fraction but the terrestrial transported flux.

p1.l18: For the most part, every fraction of terrestrial leached C is missed in previous studies, not only groundwater leaching, leading to overestimation of sink capacity of land (Jackson, Banner, & Jobbágy, 2002; Janssens et al., 2003)

p2.l2: Which fraction of DOC you are talking about? Leached or soil solution?

p2.l5-7: Needs reference.

p2.l8: Not correct. Kalbitz found strong or positive influence of pH and C:N on DOC concentration and no trend/influence on C leaching flux

p2.l11: concentration in soil or leaching flux? Moreover, soil DOC concentration changes within depth regardless of transporting period due to organic matter availability within different soil column (Jobbágy & Jackson, 2000).

p3.l7: remove “on”

p3.l17: Which classification was used for your final modelling?

p3.l23:What do you mean by SI1? give a right address to files in that folder

p3.l25: Subsoil DOC concentration cannot be calculated based on topsoil concentration using only simplified “soil class-dependent decay coefficient”. The subsoil DOC concentration, similar to top soil, is mainly controlled by total available SOC not soil class (Jobbágy & Jackson, 2000).

p3.l34: Provide the R script which was used for data analysis

p4.l4: How many sites at the end were used for modelling at the end?

p4.l21: 40 collected samples would be enough for developing a process-based model.

p4.l25: Explain why you exclude the Histosols

p4.l27: Explain the reason for the decreasing DOC concentration with depth, e.g. the

top soil concentration is controlled mainly by production, decomposition and leaching of DOC while subsoil concentration is controlled mainly by advection, diffusion and leaching.

p5.l15: The production of DOC and thus its concentration is controlled by factors such as temperature, C:N ratio, vegetation cover, soil moisture and microbial decomposition. I do not see in your model any of these factors directly applied.

p5.l20: Not true. You could use a global data product, for instance for SOC and pH for the points where you do not have the reported values. You cannot omit these parameters when it comes to representation of soil DOC

p5.l22: As I say, you cannot just ignore the soil properties which are directly controlling DOC processes and flux when global products are available that could be used.

p5.l24: soil class cannot solely represent all the physical and chemical characteristics of conditions which influence the soil DOC concentration. You must include environmental parameters such as soil moisture (DeLuca, 1992; Kalbitz, 2000; Lundquist, 1999; Michalzik, 2001), temperature (Michalzik, 1999; Moore, 2008; Raymond, 2010), pH (Fröberg, 2011; Scheel, 2008), C:N and N effect (Gödde, 1996; Kindler et al., 2011) and soil texture (Davidson et al., 2006; Filip, 1971; Sollins, Homann, & Caldwell, 1996; Stotzky, 1967; Vogel et al., 2015) to have a realistic representation of DOC.

p5.l26: You can find HWSD product which reports SOC directly (Nachtergaele et al., 2010).

p5.l26: No you cannot simply represent temperature and moisture condition by climate zones. You have temperature in your data set. Why not use that as a model parameter? and use global products for soil moisture and missing temperature data.

p5.l30: What do you mean by testing? it is not explained in the method. However, you used a dataset from 1961 to 1990 to represent a DOC concentration until year 2000? how did you fill the data from 1990 to 2000?

p6.l2: As SOC is the main source of soil DOC, all these patterns could be simply explained by SOC distribution in different biomes studied by Jobbágy and Jackson (2000).

p6.l5-11: This belongs to method

p6.l7: You should be careful to not include the above ground or litter DOC measurements in top soil DOC measurements.

p6.l16: The warmer regions, since higher temperature increases the decomposition of DOC, would exhibit lower concentration. But this would not be true in all the cases as the production of DOC can also increase during high temperature (Michalzik et al., 1999) and leaching of DOC out of soil will decrease due to the higher evapotranspiration and reduced soil water (Raymond & Saiers, 2010), resulting in an increase of DOC concentration in some regions. Hence, the authors' model based on climate zones is not valid.

p6.l26: There are many global or regional datasets that you can use for soil texture, SOC and pH.

p6.l27: Where are the results for this statement?

p6.l30: Again, where are the results for this?

p6.l31: First of all this should be in method not result. Secondly, as I mentioned above, you cannot represent correctly the processes which are influencing the soil DOC by these oversimplified factors. I suggest reconsidering your approach. As I see your results Fig.7, your model for topsoil is not capable of capturing properly the measurements at all, low concentration modelled for high measured points and vice versa.

p7. This whole page is poorly written. Back-and-forth between method and some pieces of results, with scattered arguments to support poorly constrained results.

p8. The whole same story defined for the "Top soil" section.

p8.l16:32: This all belongs to method not results. However, I am not satisfied that the subsoil concentration can be represented by only a simplified “soil class-dependent decay coefficient” which is not also well explained in this manuscript.

p9. “Application and perspective”: There are more parameters that should be included in your model as mentioned above.

p10.l7: No you cannot simply apply the water flux to the soil DOC concentration and get the leaching of DOC as the DOC removal from soil column applies the changes to the other processes involved in production/decomposition of soil DOC, resulting in change of concentration in soil as well.

Reference:

Camino-Serrano, M., Gielen, B., Luysaert, S., Ciais, P., Vicca, S., Guenet, B., ... Janssens, I. (2014). Linking variability in soil solution dissolved organic carbon to climate, soil type, and vegetation type. Global Biogeochemical Cycles, (September 2015), 497–509. https://doi.org/10.1002/2013GB004726.Received

Davidson, E. A., Janssens, I. A., Marks, D., Murdock, M., Ahl, R. S., Woods, S. W., ... Loffler, J. (2006). Temperature sensitivity of soil carbon decomposition and feedbacks to climate change. Nature, 440(7081), 165–173. https://doi.org/10.1038/nature04514

DeLuca, T. H., Keeney, D. R., & McCarty, G. W. (1992). Effect of freeze-thaw-events on mineralization of soil nitrogen. Biol. Fertil. Soils, 14, 116–120. https://doi.org/10.1007/BF00336260

Filip, Z. (1971). Clay Minerals as a Factor Influencing the Biochemical Activity of Soil Microorganisms, 74.

Fröberg, M., Hansson, K., Kleja, D. B., & Alavi, G. (2011). Dissolved organic carbon and nitrogen leaching from Scots pine, Norway spruce and silver birch stands in southern Sweden. Forest Ecology and Management, 262(9), 1742–1747. https://doi.org/10.1016/j.foreco.2011.07.033

Gödde, M., David, M. B., Christ, M. J., Kaupenjohann, M., & Vance, G. F. (1996). Carbon mineralization from the forest floor under red spruce in the northeatern {USA}. Soil Biol. Biochem., 28(9), 1181–1189.

Jackson, R., Banner, J., & Jobbágy, E. (2002). Ecosystem carbon loss with woody plant invasion of grasslands. Nature, 277(July), 623–627. https://doi.org/10.1038/nature00952.

Janssens, I. A., Freibauer, A., Ciais, P., Smith, P., Nabuurs, G., Folberth, G., . . . Dolman, A. J. (2003). Europe ' s Terrestrial Biosphere Anthropogenic CO 2 Emissions. Science, 300(June), 1538–1542. https://doi.org/10.1126/science.1083592

Jobbágy, E. G., & Jackson, R. B. (2000). The vertical distribution of soil organic carbon and its relation to climate and vegetation. Ecological Applications, 10(2), 423–436. https://doi.org/10.1890/1051-0761(2000)010[0423:TVDOSO]2.0.CO;2

Kalbitz, K., Solinger, S., Park, J.-H., Michalzik, B., & Matzner, E. (2000). Controls on the Dynamics of Dissolved Organic Matter in Soils A Review. Soil Science.

Kindler, R., Siemens, J., Kaiser, K., Walmsley, D. C., Bernhofer, C., Buchmann, N., . . . Kaupenjohann, M. (2011). Dissolved carbon leaching from soil is a crucial component of the net ecosystem carbon balance. Global Change Biology, 17(2), 1167–1185. https://doi.org/10.1111/j.1365-2486.2010.02282.x

Lundquist, E. J., Jackson, L. E., & Scow, K. M. (1999). Wet-dry cycles affect dissolved organic carbon in two California agricultural soils. Soil Biology and Biochemistry, 31(7), 1031–1038. https://doi.org/10.1016/S0038-0717(99)00017-6

Michalzik, B., Kalbitz, K., Park, J., Solinger, S., & Matzner, E. (2001). Fluxes and concentrations of dissolved organic carbon and nitrogen–a synthesis for temperate forests. Biogeochemistry, 52, 173–205. Retrieved from http://link.springer.com/article/10.1023/A:1006441620810

Michalzik, B., Michalzik, B., Matzner, E., & Matzner, E. (1999). Dynamics of dissolved organic nitrogen and carbon in a Central European Norway spruce .... European Journal of Soil Science, (December), 579–590. https://doi.org/10.1046/j.1365-2389.1999.00267.x

Moore, T. R., Paré, D., & Boutin, R. (2008). Production of dissolved organic carbon in Canadian forest soils. Ecosystems, 11(5), 740–751. https://doi.org/10.1007/s10021-008-9156-x

Nachtergaele, F., Velthuizen, H. van, Verelst, L., Batjes, N. H., Dijkshoorn, K., Engelen, V. W. P. van, ... Montanarela, L. (2010). The Harmonized World Soil Database. Proceedings of the 19th World Congress of Soil Science, Soil Solutions for a Changing World, Brisbane, Australia, 1-6 August 2010, 34–37. https://doi.org/3123

Raymond, P. A., & Saiers, J. E. (2010). Event controlled DOC export from forested watersheds. Biogeochemistry, 100(1), 197–209. https://doi.org/10.1007/s10533-010-9416-7

Scheel, T., Jansen, B., Van Wijk, A. J., Verstraten, J. M., & Kalbitz, K. (2008). Stabilization of dissolved organic matter by aluminium: A toxic effect or stabilization through precipitation? European Journal of Soil Science, 59(6), 1122–1132. https://doi.org/10.1111/j.1365-2389.2008.01074.x

Sollins, P., Homann, P., & Caldwell, B. a. (1996). Stabilization and destabilization of soil organic matter1.pdf. Geoderma, 74(1–2), 65–105. https://doi.org/10.1016/S0016-7061(96)00036-5

Stotzky, G. (1967). DIVISION OF ENVIRONMENTAL SCIENCES: CLAY MINERALS AND MICROBIAL ECOLOGY*,†. Transactions of the New York Academy of Sciences, 30(1 Series II), 11–21. https://doi.org/10.1111/j.2164-0947.1967.tb02449.x

Vogel, C., Heister, K., Buegger, F., Tanuwidjaja, I., Haug, S., Schloter, M., & Kögel-Knabner, I. (2015). Clay mineral composition modifies decomposition and sequestration of organic carbon and nitrogen in fine soil fractions. Biology and Fertility of Soils,

51(4), 427–442. https://doi.org/10.1007/s00374-014-0987-7

---

## Author Comment (AC1) · 2 Sep 2019

**Reaction to interactive comment, received and published 1 July 2019, on**
"Global database and model on dissolved carbon in soil solution", by
Langeveld, J., Bouwman, A. F., van Hoek, W. J., Vilmin, L., Beusen, A. H. W., Mogollón, J. M., and Middelburg, J. J.

We thank Zeli Tan for his reaction and constructive feedback. Below, we repeat the text of the referee in italics, followed by our response (in normal font).

*It is a very interesting study and fits well to the scope of BG. I am not an expert on soil data and thus waiting for the comments from reviewers on the validity and robustness of the method. I have one suggestion from the modelling perspective. To date, many Earth system and land surface models are using SOC as a proxy to calculate the DOC concentration in the soil and then using it with water flow to calculate DOC fluxes to inland waters. Could the authors calculate the correlation between the modeled global DOC in this study and global SOC data (such as HWSD SOC) to see how the previous proxy-based method might bias DOC flux estimates?*

Thank you for your nice suggestion and your interest in our study. Indeed, SOC is used in studies as a proxy for DOC. In addition to the sampled data from studies in the database, we also extracted ISRIC soil data from the HWSD (Batjes, 2015, 2016) for the corresponding grid cell of every database site. We included these 30 second-resolution data, as well as the dominant and mean for aggregated data to 30 minutes, in the topsoil analysis. However, the correlations for a (simple) single regression of every extracted soil parameter vs DOC where poor. For example for SOC, the Pearson correlation coefficient was not higher than 0.07 for the three datasets. In general, the mean for 30 minute grid cells yielded best results. Furthermore, in the multi-regression analysis none of the extracted soil properties had a clear added predictive value (page 6, line 28/29). Therefore, the extracted soil parameters were not used as a parameter in the model.

In response to your question, we conducted an analysis of the mentioned HWSD data for SOC topsoil (D1 in the WISE30 dataset) vs the modelled DOC concentrations (Figure 1). Correlation however is poor.

[Figure]

*Figure 1: Soil organic carbon (SOC) data (kg\*m$^{-2}$, for 0-20 cm) extracted from HWSD-WISE30sec (Batjes, 2016) plot vs. modelled DOC concentrations (mg\*C L$^{-1}$) for topsoil. SOC data are mean-value aggregated to 30 minutes. DOC*

*concentrations are the same data as in figure 7 in the article. Three extreme values with a high leverage point (SOC>200,DOC<25) were excluded. Simple linear regression coefficient of determination, $R^2 = 0.03$.*

**References**

Batjes, N. H. (2015) *World soil property estimates for broad-scale modelling (WISE30sec)*. ISRIC-World Soil Information.

Batjes, N. H. (2016) 'Harmonized soil property values for broad-scale modelling (WISE30sec) with estimates of global soil carbon stocks', *Geoderma*. Elsevier, 269, pp. 61–68.

---

## Author Comment (AC2) · 2 Sep 2019

**Reaction to interactive comment, received and published 15 July 2019, on**
"Global database and model on dissolved carbon in soil solution", by
Langeveld, J., Bouwman, A. F., van Hoek, W. J., Vilmin, L., Beusen, A. H. W., Mogollón, J. M., and Middelburg, J. J.

We thank the referee for his/her reaction and constructive feedback. The comments will help to improve our paper, especially the suggestion to also use the DOC flux data. Below, we repeat the referee's comments and questions in italics, followed by our response (in normal font). Statements about corrections in the manuscript are printed in bold.

*1. The authors have done a lot of work to assemble the data. The database will be a valuable resource. It is disappointing that the flux data are not analysed, since these are arguably more important than concentrations in the context of the carbon cycle. It would be of great interest to know the distribution and statistics of the flux data. I do not think the justification for analysing [DOC] only (page 4, line 22) is convincing. The authors say that fluxes can be obtained as the products of concentration and water flux. What happens if you do this, how do the results compare with the direct flux data? Looking at Figure 5, average fluxes might be > 20 gC/m2/a, which is about twice the value that Buckingham et al reported for non-peat UK soils.*

Thank you for your feedback. Indeed, we chose to select only the annual average DOC concentration data for analysis and not the DOC flux data. This has two main reasons: 1. the global coverage of DOC concentration data is much larger as it covers all main climate zones, unlike the DOC flux data. This is important, as we aim to model on a global scale. We aimed to explain this on page 4, line 22, referring to Table 1. 2. The model described in our paper is part of the Integrated Model to Assess the Global Environment (IMAGE) Dynamic Global Nutrient Model (DGNM) (Vilmin et al., in prep.). This model is based on a hydrological framework (Van Beek, Wada and Bierkens, 2011; Sutanudjaja *et al.*, 2018) and thus requires concentrations as an input. However, we agree we could consider the choice for analysing DOC concentrations better than currently done on page 4, line 22.

The aim of our study is to constrain DOC concentrations on a *global* scale. We recognize that a global model with this resolution (30 minutes) should be used with caution when comparing it to individual site-specific data due to scale issues. The lack of good correlations when comparing DOC concentrations with soil parameters obtained from global datasets (see text and response SC1) is an example of these scale issues.

In response to the reviewer, we modelled the DOC fluxes using the database data on climate, soil class, land use, sampling depth and extracted precipitation data from IMAGE. For hydrology, with sampling years and coordinates from the database sites as input, we used the hydrological framework PCR-GLOBWB (Van Beek, Wada and Bierkens, 2011; Sutanudjaja *et al.*, 2018), which also corrects for the impact of surface cover and land use on surface runoff. This method to calculate fluxes using concentrations and hydrology has been previously used in a similar approach on dissolved nitrogen by Beusen *et al.* (2015). We calculated the fluxes depth-specific for the subsoil. This involves a range of assumptions, simplifications and averaging. Thus, because of the spatial heterogeneity of the involved factors we cannot expect a clear correlation between the modelled and measured values. Figure 1 shows the modelled and measured fluxes, with the measured and modelled fluxes in a similar range, and with literature-based fluxes roughly twice as high as the modelled fluxes for a limited number of sites. For a global model this is an acceptable result, especially when realizing that this is a comparison of average large-scale data with point source measurements based on a range of different methods to calculate DOC fluxes (e.g. often but not always correcting for loss of water through surface runoff). We also note that the modelled fluxes generally do not exceed the mentioned 20 g C/m2/y.
**In summary, based on reviewer 1 we will improve the text in two ways: 1. we will better explain the choice for using DOC concentration data; 2. we will use the flux data for validating our modelled DOC fluxes (from concentration and water flow).**

[Figure]

*Figure 1: Subsoil annual average DOC fluxes (g C m⁻² y⁻¹) as reported in the database vs modelled subsoil annual average DOC fluxes (n=114). Five isolated extreme values were excluded. Axes have the same range for comparison purposes.*

*2. Buckingham et al compared straight average and flux-weighted concentrations, and found not much difference, but this was the UK only. Is it possible for the authors to make such a comparison for their collated data? A discussion of the possible dangers of reporting straight-average annual [DOC], which I assume is what has been done, would be helpful.*

Thank you. Much of the data we collected was already reported in studies as an average over several years and as such contains both straight average and flux-weighted concentrations (e.g. annual averages compiled by (Michalzik *et al.*, 2001)). In other cases, we indeed calculated a straight average. We did not make a comparison between the two approaches. We believe it is acceptable to include both, in particular because, as the reviewer already mentioned, in studies like Buckingham *et al.* (2008), the difference is limited. Still, we acknowledge that it is important to mention the existence of the different approaches. In conclusion, **a few sentences on this issue will be added in the manuscript. Furthermore, we will consider to include it as an additional option in a future update of the database.**

*3. The model is unimpressive – the predictions do not vary much and so there is strong bias, predictions being too high at low [DOC] and too low at high [DOC]. A problem might arise if high soil moisture promotes DOC production (positive effect) while associated high water throughput causes dilution (negative effect). Again, it would be of interest to know the implications of the modelled data for flux estimation.*

For the implications of the modelled data for flux estimation, we refer to comment 1. Although the model is far from perfect, it is the first of its kind at the global scale. We agree with the reviewer's comment on biases, and **will extend our discussion of biases on page 7, line 6 on problems associated with soil moisture and dilution**

*4. Can the different averages in Figure 4 be compared for statistical significance?*

This is a useful suggestion. We did this analysis, but it was not included in the figure. **We will add this to Figure 4 in the manuscript.**

*5. Page 4, line 20. I do not understand this argument – surely there is more decomposition of organic matter near to the soil surface, not with increasing depth? Is it more to do with escape of CO2 as gas, and the dissolved DIC flux, becoming less efficient with depth?*

We totally agree (actually, we currently build a DIC model based on this principle). **We will improve this sentence.**

*6. The paper is not that well written, it could do with a careful edit for language and grammar. The word "equatorial" is mis-spelled as "equitorial" throughout.*

**We will carefully edit the manuscript for language and grammar. We corrected the word 'equitorial' which occurred in Table 3.**

**References**

Van Beek, L. P. H., Wada, Y. and Bierkens, M. F. P. (2011) 'Global monthly water stress: 1. Water balance and water availability', *Water Resources Research*, 47(7).

Beusen, A. H. W. *et al.* (2015) 'Coupling global models for hydrology and nutrient loading to simulate nitrogen and phosphorus retention in surface water–description of IMAGE–GNM and analysis of performance', *Geoscientific model development*, 8(12), pp. 4045–4067.

Buckingham, S., Tipping, E. and Hamilton-Taylor, J. (2008) 'Dissolved organic carbon in soil solutions: a comparison of collection methods', *Soil use and management*. Wiley Online Library, 24(1), pp. 29–36.

Michalzik, B. *et al.* (2001) 'Fluxes and concentrations of dissolved organic carbon and nitrogen–a synthesis for temperate forests', *Biogeochemistry*. Springer, 52(2), pp. 173–205.

Sutanudjaja, E. H. *et al.* (2018) 'PCR-GLOBWB 2: a 5 arcmin global hydrological and water resources model', *Geoscientific Model Development*. European Geosciences Union (EGU); Copernicus, 11(6), pp. 2429–2453.

---

## Author Comment (AC3) · 2 Sep 2019

**Response to interactive comment, received and published 20 July 2019, on**
"Global database and model on dissolved carbon in soil solution", by
Langeveld, J., Bouwman, A. F., van Hoek, W. J., Vilmin, L., Beusen, A. H. W., Mogollón, J. M., and
Middelburg, J. J.

We thank the referee for his/her reaction and constructive feedback, which will lead to considerable
improvements of our paper. Below, we repeat the text of the referee in italics, followed by our
response (in normal font). Proposed revisions and corrections of the manuscript are in bold.

Consistent with the review, we first address the major comments (in which many of the other
comments are summarized) and subsequently address the minor comments. We numbered the
comments for clarification.

*Major comments:*
*1. The main problems that I see with this manuscript are as follows: Authors constructed a model*
*based on oversimplified parameters which are not fully representative of the processes that are*
*controlling soil DOC.*

Thank you very much for your comment, which we feel summarizes many of the comments given
further below. This comment is actually related to three issues which will be discussed consecutively:
(i) general aspects of global-scale modelling; (ii) representativeness of our database; (iii) scale
problems of the data.

I.    Firstly, our aim is to develop a model that accounts for the spatial variability of the main controls
      of DOC in soils on the global scale using landscapes as a basis, not site-specific. We concentrate
      on DOC concentrations aggregated to annual timescales. A general aspect is that on such
      temporal scales integrated C cycle fluxes may be strongly related to "average" biophysical
      conditions like in many other large-scale models such as Century and LPJ. This makes empirical
      relationships between DOC concentrations and environmental conditions useful for bridging the
      gap between site and landscape scales. The model we developed is not perfect, but it is the first
      of its kind at the global scale. It uses " average" conditions because in the various studies DOC
      concentrations differ within climate zones (Litaor, 1988; Tipping *et al.*, 1999), land use types
      (Aber *et al.*, 1989; Currie and Aber, 1997; Fernández-Sanjurjo, Vega and Garcia-Rodeja, 1997;
      Chantigny, 2003) and soil classes (Easthouse *et al.*, 1992; Tipping *et al.*, 1999; Johnson *et al.*,
      2006; Don and Schulze, 2008; Harrison *et al.*, 2008). In that respect our model is consistent with
      other global models that use biome type as a basis for modelling global soil respiration (Raich
      and Schlesinger, 1992) or climatic life zones for estimating global soil organic carbon pools (Post
      *et al.*, 1982).

II.   A prerequisite for using empirical relationships to calculate DOC concentrations is to have data
      which sufficiently cover the heterogeneity of environmental conditions occurring in natural and
      agricultural landscapes. Because at this scale "average" biophysical conditions are more useful,
      the parameters we use in our final model are not directly representative of the processes
      controlling soil DOC. In our study, we considered many of the parameters the reviewer
      mentioned and included those in the database when available in the literature. A problem we
      encountered in the database analysis, when building the (topsoil) model, is that many
      parameters are only reported for a limited part of the data entries, which we describe in the
      chapter 'General aspects and data analysis' on page 5, line 16-21:
              *"The distribution of data on potential drivers of DOC concentrations is unbalanced; the*
      *choice of factors included in sampling studies varies (Table 2). This is a problem earlier recognized*
      *on a smaller scale (Kalbitz et al., 2000; Evans, Monteith and Cooper, 2005). As a result of these*

*data gaps, for many factors, analysis is only possible on a limited part of the DOC concentration data. Moreover, including a few factors strongly reduces the amount of data involved (e.g. for topsoils, including pH, SOC and CN cuts the amount of data from 255 to 40), thereby impeding analysis at a global scale. Indeed, significant relations can be found for sub-sets of the data."*

For example, for data exclusively from Spodosols (analysis including all soil types gave no significant results), a (relatively weak) relation can be found for C/ N ratio vs. DOC concentrations (Figure 1). However, a model like this would only depend on a limited amount of data and is only valid for this soil type, based on data almost exclusively found in temperate forests. Such a model does not fit with the objective of our study.

[Figure]

*Figure 1: Single regression of C/N ratio vs DOC concentrations from the database, for only data from the Spodosol soil class. n=18, from polar(1), continental (2) and temperate zones (16). One high leverage point outlier was excluded. Lab incubation and B-horizon data were excluded, similar to the analysis in the article. Data cover mainly forests.*

We want to emphasise that the aim of our study is to constrain DOC concentrations on a *global scale*, not to construct a model to estimate concentrations or fluxes in or throughout a limited region, biome, ecosystem or group of point sources. Therefore, we can only include parameters or model factors that are available on *global scale*, i.e. for all (or at least a large part) of the data in the database. This was not the case for many of the potential drivers reported in the literature.

III. Scale issues. We agree with the reviewer that, in absence of suitable data for many entries, as an alternative soil property data like SOC can be extracted from global grids such as the HWSD. Thanks to the reviewer we noticed that our text was not clear enough that we also used global HWSD soil data extracted from grids in various ways in our regression analysis. To clarify; we extracted 19 different parameters from ISRIC soil data from the HWSD (Batjes, 2015, 2016) for the corresponding grid cell coordinates of every database site. We included these 30 second-resolution data, as well as different types of aggregations to 30 minutes, in the topsoil analysis. However, the correlations for a (simple) single regression of every extracted soil parameter vs DOC where poor. In the combined forward/backward multiple-linear regression analysis the algorithm didn't select any of the extracted variables, as the added value to the model was too low. Instead, the algorithm selected the factors that are currently in the model we presented in the manuscript.

**In conclusion: In our revision we will add a general discussion of global-scale modelling, use of average conditions and representativeness of data, and better describe how we used and tested global grid-based data to extract data for our regression analysis.**

**2.** *Additionally, they did not use a proper climate data period.*

We believe the reviewer refers to the use of data extracted from grid for precipitation and temperature. For the database entries, as an alternative to the limited data reported in literature, we used data from the climate database of the IMAGE model (Stehfest *et al.*, 2014), which involves the CRU climate data. Though we included the correct climate data period we noticed, thanks to the reviewer, that we used an outdated reference for these climate data, **which we will correct by replacing (New, Hulme and Jones, 1997) by (Harris *et al.*, 2013).**

In addition we want to emphasize we aimed to constrain and model DOC concentrations on a global scale. We therefore developed the model based on forward/backward multiple-linear regression algorithms. The resulting model was then applied on calculating a global distribution of annual average DOC concentrations. The period 1970-2000 was used as example. Naturally, for calculating DOC concentrations for the year 1970 we used climate data representative for 1970, and correspondingly for the other years. We believe this to be the appropriate method.

*3. The bright side of this work was the time that authors spent on collecting the measurements from various studies, although most of them were already reported by Camino-Serrano et al. (2014).*

Thank you for the recognition of the effort we made. We believe our database is novel compared to the one of Camino-Serrano (2014), which, though impressive, is not publicly available. Comparing both datasets, the studies analysed in the 1st part of the study by Camino-Serrano (2014) mainly cover locations in Europe, as can be viewed in Figure S1 of their supplementary information (Camino-Serrano *et al.*, 2014). More important, the majority of the measurements used there are from the International Cooperative Programme on Assessment and Monitoring of Air Pollution Effects on Forests (ICP Forests) (Nieminen *et al.*, 2016), as Camino-Serrano (2014) show in their Table 1. We, however, on purpose did not include data from ICP Forests in our database, as these data are not publicly available; availability and accessibility were a prerequisite for inclusion of a study in our database. Further, we aimed to constrain and model DOC concentrations on a *global* scale, where Camino-Serrano (2014) in the 2nd part of their study built models describing DOC concentrations in European broadleaved and coniferous forests. We therefore believe both studies complement each other.

*4. Following some major comments:*
*1) I search the author's name and paper name on PANGAEA.de and I could not find any database or model codes. It will be useful if the authors provide a direct link to such data. However, I extracted the model results from the provided netcdf file attached to this manuscript. First of all, the time dimension in the model output has 7 layers which seem almost identical to me. What are those 7 layers and why are they almost identical?. Secondly, the model output is for 1970 to 2000 while their database is not in this time period. Thirdly, the model outputs are significantly lower than their reported database and previous studies. I get average of 16.5 mg C L-1 (median:13.7) for the topsoil and average of 2.1 mg C L-1 (median: 0.9) at the subsoil layer. This shows the weakness of model in representing the processes which are controlling the DOC concentration (mentioned above).*

**The Supplementary Information, including the database, will be submitted to PANGAEA when submitting the final accepted article, so we can include the DOI in the article.**

Responding to the reviewers first question, the 7 layers in the time dimension are the 7 five-yearly grids for the period of 1970-2000. **We will add to the SI readme that we present five-yearly grids**. The 7 layers are similar, as some of the model parameters only change slowly over time, e.g. climate zones. Therefore we chose to present five-yearly grids. However, there are still significant changes in the spatial distribution over time, in particular due to land use change. Therefore we included the time-dimension in the netcdf.

In response to the reviewers second point: we refer to the answer to question 2, where we addressed this already.

Responding to the reviewers third point: when calculating the average and median of *all* cells, this would indeed be in the range of the values mentioned by the reviewer. However, the calculation of these (topsoil) mean values then also includes the many grid cells with an area cover of Aridisols (soil class), ice cover, rock cover, shifting sands, salt plains or hot deserts. For these areas, we did not use the model but assume a fixed DOC concentration equal to that in precipitation, for which we use 1.55 mg C/L, the median value from the overview by Aitkenhead-Peterson, McDowell and Neff (2003). We explain this in footnote 2 of table of Table 3. **We will add a sentence about this in the text to emphasise this.** Thus, when comparing the averages or medians of the netcdf grid data with the model in the approach as described by the reviewer, the values do not correctly represent the performance of the model. **We will revise where necessary the presentation and discussion of the model performance (e.g. in Figure 5, Figure 10, Table 3, Table 4 and the corresponding text).**

**5.**
*2) I noticed in your dataset there are many sites that you only have DOC flux reported. If you do not want to analyse this flux, all those sites should be removed from your dataset. Keeping just sites with observed DOC concentrations will leave only 550 measured points where plenty of them have same sample ID. Hence the claim of having 762 entries and 351 sites is not really true for soil DOC. Also 94 points are non-yearly averaged, with at least 40 of them are only once measured. These data should be removed from data analysis as they cannot be representative of a site. The future corrected model should be run again excluding these points and all result should be corrected.*

The database is not limited to annual average DOC concentrations. Apparently the manuscript was not clear enough so **we will improve the database description, for example in section 3.2.2**.The reviewer states that data on fluxes and non-yearly average concentrations should be excluded from the data analysis and model construction. That is exactly what we did, **so we need to improve the description in our manuscript in that respect.**

**6.**
*3) I do not see any model vs measurements validation for subsoil.*

Thank you. Indeed, we did not make a plot for the subsoil similar as done for the topsoil in Figure 7. This however is due to the different approach for the subsoil, for which we present different models for every USDA soil class (eq.2). Every single individual model of these models consists of a simple regression function through the measurement data (relative to the topsoil concentration). See Figure 10 for the example for Spodosols. As there is only one independent variable in this model, namely depth, the RMSE and $R^2$ already present the degree of model performance. Both are given in Table 4 for every single soil class model and discussed in section 3.2.2. We therefore believe that plotting the several model vs measurements graphs would be redundant.

**7.***Comments/Question:*
*p1.l15: 2.9 Pg C yr-1 is not the processed fraction but the terrestrial transported flux.*

We currently describe the C flux component as "*Global inland waters are estimated to process about 2.9 Pg C/y …*", which is in correspondence with Regnier *et al* (2013) stating that "*The present-day bulk C input (natural plus anthropogenic) to freshwaters was recently estimated at 2.7–2.9 Pg C yr⁻ ¹*"(Regnier *et al.*, 2013, p. 2). We prefer the word 'processed' above 'transport', as with respect to C,

inland waters are to be viewed as active systems (Cole *et al.*, 2007; Aufdenkampe *et al.*, 2011) and not 'passive pipes' (Cole *et al.*, 2007, pp. 174, Figure 1a).

*8. p1.l18: For the most part, every fraction of terrestrial leached C is missed in previous studies, not only groundwater leaching, leading to overestimation of sink capacity of land (Jackson, Banner, & Jobbágy, 2002; Janssens et al., 2003)*

We agree with the reviewer and notice the current phrase could be interpreted as if we do not. **We will modify the sentence.**

*9. p2.l2: Which fraction of DOC you are talking about? Leached or soil solution?*

We mean DOC in soil solution and **will add this in the manuscript.**

*10. p2.l5-7: Needs reference.*

In Table 1 we summarized the many possible drivers of DOC concentrations and the studies they are reported in. In this table, we refer both to studies that did and those that did *not* identify a factor as a main control. Though we refer to Table 1 one sentence later, **we will also refer to Table 1 after this sentence.**

*11. p2.l8: Not correct. Kalbitz found strong or positive influence of pH and C:N on DOC concentration and no trend/influence on C leaching flux*

We acknowledge that 'leaching' in here might be interpreted as specifically the flux, where we mean 'solution'. **We will clarify this.**

*12. p2.l11: concentration in soil or leaching flux? Moreover, soil DOC concentration changes within depth regardless of transporting period due to organic matter availability within different soil column (Jobbágy & Jackson, 2000).*

We believe that 'dissolved C in soil pore water' in here is a clear definition, as in these sentences we explain the boundaries of this study.

*13. p3.l7: remove "on"*

**We will address this during a careful edit for language and grammar.**

*14. p3.l17: Which classification was used for your final modelling?*

We used the alternative classification following Batjes (2015, 2016), as we state in the first sentence of the topsoil database analysis. To make this more clear, **we will therefore move this sentence to the beginning of section 3.1, where we start the analysis of the database.**

*15. p3.l23:What do you mean by SI1? give a right address to files in that folder*

SI is the abbreviation for Supplementary Information. So, SI1 means number 1 of the SI. **We will explain the full term in the text.** We have also given a clear description of all Supplementary Information in the readme file in the SI.

*16. p3.l25: Subsoil DOC concentration cannot be calculated based on topsoil concentration using only simplified "soil class-dependent decay coefficient". The subsoil DOC concentration, similar to top soil, is mainly controlled by total available SOC not soil class (Jobbágy & Jackson, 2000).*

The aim of our study is to constrain DOC concentrations on a global scale, not to construct a model to estimate concentrations or fluxes in or throughout a limited region, biome, ecosystem or group of point sources. We totally acknowledge and agree that the parameters we use in our final model are not fully representative of all the processes controlling soil DOC. We however believe our current approach is valid for making a first estimate on the global scale. We explained this further in question 1.

*17. p3.l34: Provide the R script which was used for data analysis*

**We will add the scripts for the (multi-)regression analysis for both top- and subsoil.**

*18. p4.l4: How many sites at the end were used for modelling at the end?*

We addressed this in the respective chapters for topsoil (3.2) and subsoil (3.3), specifically in Figure 7 and Table 4. Still, we acknowledge we should also mention it in the text, **so we will some text on this.**

*19. p4.l21: 40 collected samples would be enough for developing a process-based model.*

The aim of our study is to constrain the dissolved C concentrations on a global scale, not to construct a model to estimate concentrations or fluxes in or throughout a limited region, biome, ecosystem or group of point sources. As we show in Table 1, the global cover of DIC concentration is limited. In a following study we are currently working on, we will also include a model for DIC, using a different approach.

*20. p4.l25: Explain why you exclude the Histosols*

We currently explain this later in the text, but **will also briefly explain it here.**

*21. p4.l27: Explain the reason for the decreasing DOC concentration with depth, e.g. the top soil concentration is controlled mainly by production, decomposition and leaching of DOC while subsoil concentration is controlled mainly by advection, diffusion and leaching.*

Thank you for the suggestion. We believe we do this already in the introduction, in particular in the first two paragraphs on page 2. **We will however also include 'advection, diffusion and leaching' in the text there**.

*22. p5.l15: The production of DOC and thus its concentration is controlled by factors such as temperature, C:N ratio, vegetation cover, soil moisture and microbial decomposition. I do not see in your model any of these factors directly applied.*

The aim of our study is to constrain DOC concentrations on a global scale, not to construct a model to estimate concentrations or fluxes in or throughout a limited region, biome, ecosystem or group of point sources. We totally acknowledge and agree that the parameters we use in our final model are not fully representative of all the processes controlling soil DOC. We however believe our current approach is valid for making a first estimate on the global scale. We addressed this point in the answer to question 1.

*23. p5.l20: Not true. You could use a global data product, for instance for SOC and pH for the points where you do not have the reported values. You cannot omit these parameters when it comes to representation of soil DOC*

We believe it to be valid approach, as we *did* include global products in our analysis. We addressed this point in the response to question 1.

*24. p5.l22: As I say, you cannot just ignore the soil properties which are directly controlling DOC processes and flux when global products are available that could be used.*

We totally acknowledge and agree that the parameters we use in our final model are not direct controls of soil DOC which at this scale are related to "average" conditions (see response #1)

*25. p5.l24: soil class cannot solely represent all the physical and chemical characteristics of conditions which influence the soil DOC concentration. You must include environmental parameters such as soil moisture (DeLuca, 1992; Kalbitz, 2000; Lundquist, 1999; Michalzik, 2001), temperature (Michalzik, 1999; Moore, 2008; Raymond, 2010), pH (Fröberg, 2011; Scheel, 2008), C:N and N effect (Gödde, 1996; Kindler et al., 2011) and soil texture (Davidson et al., 2006; Filip, 1971; Sollins, Homann, & Caldwell, 1996; Stotzky, 1967; Vogel et al., 2015) to have a realistic representation of DOC.*

See our response #1.

*26. p5.l26: You can find HWSD product which reports SOC directly (Nachtergaele et al., 2010).*

We used global products in our analysis. We addressed this point in the answer to question 1.

*27. p5.l26: No you cannot simply represent temperature and moisture condition by climate zones. You have temperature in your data set. Why not use that as a model parameter? and use global products for soil moisture and missing temperature data.*

We totally acknowledge and agree that the parameters we use in our final model are not fully representative of all the processes controlling soil DOC. We did include global products in our analysis. Temperature was not selected by the algorithm in the combined forward/backward multi-regression analysis. See response #1.

**28.** *p5.l30: What do you mean by testing? it is not explained in the method. However, you used a dataset from 1961 to 1990 to represent a DOC concentration until year 2000? how did you fill the data from 1990 to 2000?*

We explain it in the sentence that follows on page 5 line 29-32. We agree the short sentence on 'testing' is not clear and **will edit this.** With regard to the second question; We did use the correct dataset, but unfortunately the incorrect reference. We will correct this, as we explained in the answer to question 2.

**29.** *p6.l2: As SOC is the main source of soil DOC, all these patterns could be simply explained by SOC distribution in different biomes studied by Jobbágy and Jackson (2000).*

We included SOC in the regression analysis (from global products), but the regression algorithm did not include any of those in the best fit (see our response #1). We believe the global relations are not to as simple as the reviewer suggests.

**30.** *p6.l5-11: This belongs to method*

In this paragraph we analyse specifically the topsoil results of the database compilation. The database compilation is the first part of our method, the database is the result of it. Therefore, we believe these lines to belong to the results and analysis section.

**31.** *p6.l7: You should be careful to not include the above ground or litter DOC measurements in top soil DOC measurements.*

We included no aboveground data in the database. The database contains three entries which specifically are referred to as 'litter layer' in their sources (Markewitz *et al.*, 2004; Gielen *et al.*, 2011). According to the soil sampling guidelines of both the USDA (Schoeneberger, Wysocki and Benham, 2012, p. 6) and the FAO (Jahn *et al.*, 2006, p. 32) a litter layer that has begun to decompose could be included as part of the O horizon, namely as the Oi horizon. Michalzik and Matzner (1999) also included this Oi horizon in their early overview of DOC concentrations and fluxes in different soil horizons in forests. We acknowledge however that in this matter different approaches and views exist among soil scientists. As we do not only assess the mineral soil horizons but also the O horizon, we believe it to be acceptable to include litter layer and Oi measurements in the topsoil DOC data.

**32.** *p6.l16: The warmer regions, since higher temperature increases the decomposition of DOC, would exhibit lower concentration. But this would not be true in all the cases as the production of DOC can also increase during high temperature (Michalzik et al., 1999) and leaching of DOC out of soil will decrease due to the higher evapotranspiration and reduced soil water (Raymond & Saiers, 2010), resulting in an increase of DOC concentration in some regions. Hence, the authors' model based on climate zones is not valid.*

We agree that drivers can have differing or even opposing effects depending on the location, like we explain on page 7, line 26-32. Also, some effects might be indirect, like in the example we give on page 5, line 14-15. We however believe our current approach is valid for making a first estimate on the global scale. See our response #1.

**33.** *p6.l26: There are many global or regional datasets that you can use for soil texture, SOC and pH.*

We did include global products in our analysis. However they were not selected in the combined forward/backward multiple-linear regression analysis by the algorithm. We explained this further in the response to question 1.

**34.** *p6.l27: Where are the results for this statement?*

Indeed, in order to keep focus we did not add the many non-significant results. However, we understand the interest of the reviewer in these results. **In addition to the analysis code, we will add the extracted data for all database entries (both the 30 second resolution and aggregations to 30 minutes, representing the mean and dominant value) in a separate sheet in the database, so the analysis can be reproduced.**

**35.** *p6.l30: Again, where are the results for this?*

See #34.

**36.** *p6.l31: First of all this should be in method not result. Secondly, as I mentioned above, you cannot represent correctly the processes which are influencing the soil DOC by these oversimplified factors. I suggest reconsidering your approach. As I see your results Fig.7, your model for topsoil is not capable of capturing properly the measurements at all, low concentration modelled for high measured points and vice versa. p7. This whole page is poorly written. Back-and-forth between method and some pieces of results, with scattered arguments to support poorly constrained results.*

With regard to the reviewers first point, we believe this part should belong in the 'results and analysis' part of the manuscript, as this paragraph (page 6 line 25 – page 7 line 5) presents the results of the multi-regression analysis. The model itself is a result of our study, not merely a method.

With regard to the reviewers second point, we would like to point out that the aim of our study is to constrain DOC concentrations on a global scale, not to construct a model to estimate concentrations or fluxes in or throughout a limited region, biome, ecosystem or group of point sources. We totally acknowledge and agree that the parameters we use in our final model are not fully representative of all the processes controlling soil DOC. We however believe our current approach is valid for making a first estimate on the global scale. We addressed this point in the answer to question 1.

Finally, we will address the composition of this chapter during **a careful edit for language and grammar.** We however wish to emphasise that the product of the model analysis is on purpose presented as a result, and not merely a method, as the model is the final outcome of our study.

**37.** *p8. The whole same story defined for the "Top soil" section.*

See # 36.

**38.** *p8.l16:32: This all belongs to method not results. However, I am not satisfied that the subsoil concentration can be represented by only a simplified "soil class-dependent decay coefficient" which is not also well explained in this manuscript.*

We addressed this in our reply to question #36, where, with regard to the use of soil class as a factor, we also refer to our extensive answer to comment #1.

**39.** *p9. "Application and perspective": There are more parameters that should be included in your model as mentioned above.*

We would like to point out that the aim of our study is to constrain DOC concentrations on a global scale, not to construct a model to estimate concentrations or fluxes in or throughout a limited region, biome, ecosystem or group of point sources. We totally acknowledge and agree that the parameters we use in our final model are not fully representative of all the processes controlling soil DOC. We however believe our current approach is valid for making a first estimate on the global scale. We addressed this point in the answer to comment #1.

**40.** *p10.l7: No you cannot simply apply the water flux to the soil DOC concentration and get the leaching of DOC as the DOC removal from soil column applies the changes to the other processes involved in production/decomposition of soil DOC, resulting in change of concentration in soil as well.*

We agree with the reviewer that the water flux should not be multiplied with the DOC concentration at a certain depth *'x'* to calculate the flux over the total column. We believe we are neither stating that. Though not the aim of this study, we would consider a method in which, *for annual average values*, the DOC flux at depth *x* can be gained by multiplying the concentration at depth *x* with the water flux at depth *x*. See also our answer on question 1 of reviewer 1.

**References**

Aber, J. D. *et al.* (1989) 'Nitrogen saturation in northern forest ecosystems.', *BioScience*, 39(6), pp. 286–378.

Aitkenhead-Peterson, J. A., McDowell, W. H. and Neff, J. C. (2003) 'Sources, production, and regulation of allochthonous dissolved organic matter inputs to surface waters', in *Aquatic Ecosystems*. Elsevier, pp. 25–70.

Aufdenkampe, A. K. *et al.* (2011) 'Riverine coupling of biogeochemical cycles between land, oceans, and atmosphere', *Frontiers in Ecology and the Environment*. Ecological Society of America, 9(1), pp. 53–60.

Batjes, N. H. (2015) *World soil property estimates for broad-scale modelling (WISE30sec)*. ISRIC-World Soil Information.

Batjes, N. H. (2016) 'Harmonized soil property values for broad-scale modelling (WISE30sec) with estimates of global soil carbon stocks', *Geoderma*. Elsevier, 269, pp. 61–68.

Camino-Serrano, M. *et al.* (2014) 'Linking variability in soil solution dissolved organic carbon to climate, soil type, and vegetation type', *Global Biogeochemical Cycles*. Wiley Online Library, 28(5), pp. 497–509.

Chantigny, M. H. (2003) 'Dissolved and water-extractable organic matter in soils: a review on the influence of land use and management practices', *Geoderma*. Elsevier, 113(3–4), pp. 357–380.

Cole, J. J. *et al.* (2007) 'Plumbing the global carbon cycle: integrating inland waters into the terrestrial carbon budget', *Ecosystems*. Springer-Verlag, 10(1), pp. 172–185. doi: 10.1007/s10021-006-9013-8.

Currie, W. S. and Aber, J. D. (1997) 'Modeling leaching as a decomposition process in humid montane forests', *Ecology*. Wiley Online Library, 78(6), pp. 1844–1860.

Don, A. and Schulze, E.-D. (2008) 'Controls on fluxes and export of dissolved organic carbon in grasslands with contrasting soil types', *Biogeochemistry*. Springer, 91(2–3), pp. 117–131.

Easthouse, K. B. *et al.* (1992) 'Dissolved organic carbon fractions in soil and stream water during variable hydrological conditions at Birkenes, southern Norway', *Water resources research*. Wiley Online Library, 28(6), pp. 1585–1596.

Evans, C. D., Monteith, D. T. and Cooper, D. M. (2005) 'Long-term increases in surface water dissolved organic carbon: observations, possible causes and environmental impacts', *Environmental pollution*. Elsevier, 137(1), pp. 55–71.

Fernández-Sanjurjo, M. J., Vega, V. F. and Garcia-Rodeja, E. (1997) 'Atmospheric deposition and ionic concentration in soils under pine and deciduous forests in the river Sor catchment (Galicia, NW Spain)', *Science of the total environment*. Elsevier, 204(2), pp. 125–134.

Gielen, B. *et al.* (2011) 'The importance of dissolved organic carbon fluxes for the carbon balance of a temperate Scots pine forest', *Agricultural and Forest Meteorology*. Elsevier, 151(3), pp. 270–278.

Harris, I. *et al.* (2013) 'Updated high-resolution grids of monthly climatic observations–the CRU TS3. 10 Dataset', *International journal of climatology*. Wiley Online Library, 34(3), pp. 623–642.

Harrison, A. F. *et al.* (2008) 'Potential effects of climate change on DOC release from three different soil types on the Northern Pennines UK: examination using field manipulation experiments', *Global Change Biology*. Wiley Online Library, 14(3), pp. 687–702.

Jahn, R. *et al.* (2006) *Guidelines for soil description*. FAO.

Johnson, M. S. *et al.* (2006) 'DOC and DIC in flowpaths of Amazonian headwater catchments with hydrologically contrasting soils', *Biogeochemistry*. Springer, 81(1), pp. 45–57.

Kalbitz, K. *et al.* (2000) 'Controls on the dynamics of dissolved organic matter in soils: a review', *Soil science*. LWW, 165(4), pp. 277–304.

Litaor, M. I. (1988) 'Soil solution chemistry in an alpine watershed, Front Range, Colorado, USA', *Arctic and Alpine Research*. Taylor & Francis, 20(4), pp. 485–491.

Markewitz, D. *et al.* (2004) 'Nutrient loss and redistribution after forest clearing on a highly weathered soil in Amazonia', *Ecological Applications*. Wiley Online Library, 14(sp4), pp. 177–199.

Michalzik, B. and Matzner, E. (1999) 'Dynamics of dissolved organic nitrogen and carbon in a Central European Norway spruce ecosystem', *European journal of soil science*. Wiley Online Library, 50(4), pp. 579–590.

New, M., Hulme, M. and Jones, P. (1997) 'A 1961–1990 mean monthly climatology of global land areas', *Climatic Research Unit, University of East Anglia, Norwich, UK*.

Nieminen, T. *et al.* (2016) *Soil Solution Collection and Analysis, ICP Forests Manual*. Eberswalde, Germany. Available at: https://www.icp-forests.org/pdf/manual/2016/ICP_Manual_2016_01_part11.pdf.

Post, W. M. *et al.* (1982) 'Soil carbon pools and world life zones', *Nature*. Nature Publishing Group, 298(5870), p. 156.

Raich, J. W. and Schlesinger, W. H. (1992) 'The global carbon dioxide flux in soil respiration and its relationship to vegetation and climate', *Tellus B*. Wiley Online Library, 44(2), pp. 81–99.

Regnier, P. *et al.* (2013) 'Anthropogenic perturbation of the carbon fluxes from land to ocean', *Nature Geoscience*. Nature Publishing Group, 6(8), pp. 597–607.

Schoeneberger, P. J., Wysocki, D. A. and Benham, E. C. (2012) *Field book for describing and sampling soils*. Government Printing Office.

Stehfest, E. *et al.* (2014) *Integrated assessment of global environmental change with IMAGE 3.0: Model description and policy applications*. Netherlands Environmental Assessment Agency (PBL).

Tipping, E. *et al.* (1999) 'Climatic influences on the leaching of dissolved organic matter from upland UK moorland soils, investigated by a field manipulation experiment', *Environment International*. Elsevier, 25(1), pp. 83–95.